# Training Hybrid Deep Quantum Neural Network for Efficient Reinforcement Learning

## Abstract

Quantum circuits embed data in a Hilbert space whose dimensionality grows exponentially with the number of qubits, allowing even shallow parameterised quantum circuits (PQCs) to represent highly-correlated probability distributions that are costly for classical networks to capture. Reinforcement-learning (RL) agents, which must reason over long-horizon, continuous-control tasks, stand to benefit from this expressive quantum feature space, but only if the quantum layers can be trained jointly with the surrounding deep-neural components. Current gradient-estimation techniques (e.g., parameter-shift rule) make such hybrid training impractical for realistic RL workloads, because every gradient step requires a prohibitive number of circuit evaluations and thus erodes the potential quantum advantage. We introduce qtDNN, a tangential surrogate that locally approximates a PQC with a small differentiable network trained on-the-fly from the same minibatch. Embedding qtDNN inside the computation graph yields scalable batch gradients while keeping the original quantum layer for inference. Building on qtDNN we design hDQNN-TD3, a hybrid deep quantum neural network for continuous-control reinforcement learning based on the TD3 architecture, which matches or exceeds state-of-the-art classical performance on popular benchmarks. The method opens a path toward applying hybrid quantum models to large-scale RL and other gradient-intensive machine-learning tasks.

## 1 Introduction

Quantum computing has attracted significant attention for its potential to address classically hard problems by exploiting Hilbert spaces that grow exponentially with the number of qubits (Biamonte et al., 2017). Machine learning (ML) is a domain that could benefit greatly, as patterns are extracted from increasingly large datasets (Vaswani et al., 2023; DeepSeek-AI et al., 2025b) and demand efficient methods for advancing AI (Robison, 2024; Samsi et al., 2023; Wells, 2025). Quantum machine learning (QML) aims to enhance learning efficiency and generalization beyond classical methods (see Table 1 in Biamonte et al. (2017)). Although fault-tolerant machines to run algorithms such as Harrow et al. (2009) or Brassard et al. (2000) are not yet available, rapid progress on noisy intermediate-scale quantum (NISQ) devices has spurred QML methods that operate under noise constraints (Wang and Liu, 2024). On one side, quantum supremacy experiments (Arute et al., 2019; Acharya et al., 2025; Gao et al., 2025) show NISQ devices can generate distributions intractable for classical hardware. On the other, reinforcement learning (RL) has become central in AI (ACM, 2025), modeling regularities in interaction data (Vaswani et al., 2023) to refine policies and achieve strong task performance (Schulman et al., 2017; Haarnoja et al., 2018; DeepSeek-AI et al., 2025a). The highly correlated distributions sampled from NISQ devices (Andersen et al., 2025) could help model complex correlations in RL (Lo et al., 2023; Huang et al., 2022; Vaswani et al., 2023), potentially improving reward, efficiency, and robustness.

Recent work has begun exploring quantum reinforcement learning (QRL). For example, Jin et al. (2025) modified PPO with parameterised quantum circuits (PQCs) and achieved performance similar to classical baselines (Raffin, 2020) on simple benchmarks (Towers et al., 2024). Hybrid deep quantum neural networks (hDQNNs) combine classical deep neural networks that map observations into real parameters, fed into PQCs operating in Hilbert space $\mathcal{H}_{N_q} \cong \mathbb{C}^{2^N}$. Sampled outputs act as bottleneck features. Such architectures show promise for high-dimensional state and action

tasks (Dong et al., 2008; Paparo et al., 2014; Dunjko et al., 2017; Jerbi et al., 2021; Wu et al., 2025). Yet a major barrier is the prohibitive cost of backpropagating through PQCs to train the surrounding classical DNNs.

## 1.1 RELATED WORK

Gradient estimation techniques for PQCs (Wang and Liu, 2024) exist but remain impractical on hardware with large input/output dimensions, as they require many sequential circuit evaluations and cannot exploit mini-batch parallelism. This limits throughput and makes hDQNN training inefficient on real NISQ devices. Previous QRL studies demonstrated feasibility on small tasks (Jin et al., 2025), but lacked scalable training strategies. Classical surrogates have been proposed to mimic PQCs, yet typically either replace inference (losing quantum benefit) or lack guarantees of gradient fidelity.

## 1.2 CONTRIBUTIONS

In this work we introduce **hDQNN-TD3**, a hybrid quantum actor-critic agent addressing the PQC back-propagation bottleneck via a new surrogate, **qtDNN**, which:

1. Provides a mini-batch-local differentiable approximation of the PQC used only in the backward pass, while the forward path and inference remain quantum.

2. Enables scalable mini-batch training by supplying gradients without additional quantum evaluations, compatible with high-throughput GPU pipelines.

3. Offers theoretical support through a local gradient-fidelity guarantee, ensuring that surrogate back-propagation perturbs upstream gradients only within controlled tolerances.

4. Demonstrates empirical effectiveness on the `Humanoid-v4` benchmark, outperforming widely adopted classical baselines (Zhao, 2024; 2023a;b) and validating the nontrivial modeling capacity of PQCs compared with fully connected, random, or zero-layer alternatives.

## 2 NEURAL PQC TANGENTIAL APPROXIMATION

### 2.1 BACK-PROPAGATION AND THE QUANTUM BOTTLENECK

Modern machine learning methods are established with back-propagation-based training: each operation in the computation graph supplies analytical Jacobians, so the loss can be differentiated w.r.t. millions of parameters and "backpropagated" through them with a single reverse pass.

On the other hand, hDQNNs are typically developed using PQCs to act as functional quantum layers (QL) without learnable parameters. The described flow is typically interrupted by a QL [1]. Fig. 1a shows the typical layout: a classical neural network **PreDNN** produces PQC control parameters $\mathbf{q}_i$, a **QL** outputting the empirical marginal vector $\mathbf{q}_o = \hat{\boldsymbol{p}} \in [0,1]^N$ after $S$ shots, and a second classical neural network **PostDNN** maps $\mathbf{q}_o$ to the final prediction. $d_{\text{c-link}}$ classical links directly connecting PreDNN and PostDNN are added on top of the QL. These direct links allow classical communication between the DNNs around the QL, to help the PostDNN properly handle the QL's sampled output.

It is however challenging to backpropagate loss efficiently through QLs. Unlike Gaussian reparameterisation layers, a PQC has no closed-form gradient. Various strategies have been developed in recent years (cf. Wang and Liu (2024)). Most methods obtain the gradient vector of a PQC with respect to its input parameters by varying each input parameter and using repeated PQC executions to estimate its partial derivative with respect to each varied input. Such methods become quickly intractable as it must run $\mathcal{O}(I)$ extra circuits per output dimension, driving the hardware cost to $I \times O \times S \times N_b$ evaluations per mini-batch for the QL with $I$ and $O$ input and output dimensions respectively (cf. App. A.1.2). Despite requiring many repeated measurements that scale with the size of input and output, methods based on direct PQC gradient estimation make it intrinsically unfeasible to batch-backpropagate in parallel, prohibiting efficient mini-batched training for a hDQNN, especially when it is applied to complex reinforcement learning.

---

[1]QL: for each input control vector $\mathbf{q}_i$, run the PQC for $S$ shots, aggregate the $S$ measured bit-strings $\{\mathbf{b}^{(s)}\}_{s=1}^{S}$, and return the per-qubit empirical marginal vector $\hat{\boldsymbol{p}}(\mathbf{q}_i) \in [0,1]^N$ with entries $\hat{p}_\ell = \frac{1}{S}\sum_{s=1}^{S} b_\ell^{(s)}$.

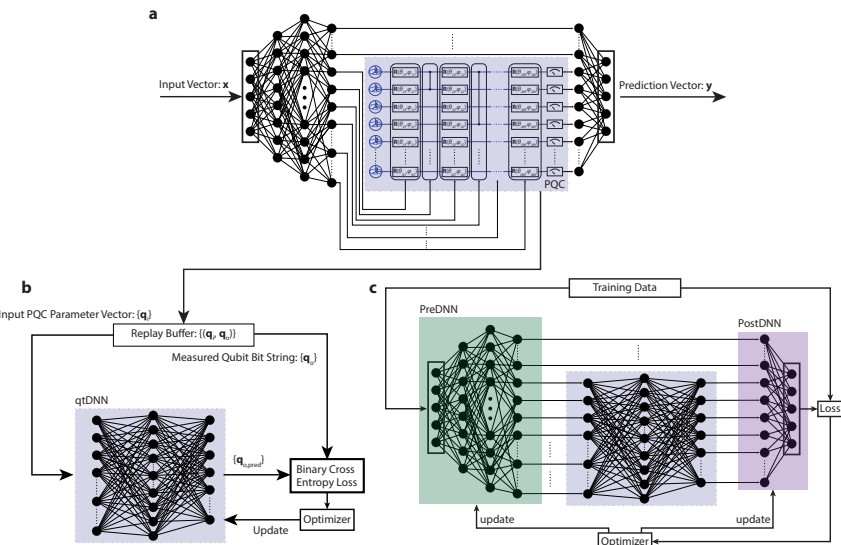

Figure 1: **a**, a typical hDQNN contains a PreDNN connected to a PostDNN via both $d_{\text{c-link}}$ direct connections and a quantum layer realised as a parametrised quantum circuit. The hDQNN inferences through the training dataset and records $(\mathbf{x}, \mathbf{y}_{\text{pred}}, \mathbf{y}, \mathbf{q}_i, \mathbf{q}_o)$ of each forward pass into a buffer $\mathcal{M}$. In an update step, a batch, $\mathcal{B}$, of $N_b$ entries is sampled from $\mathcal{M}$ for updating hDQNN parameters. **b**, $\{(\mathbf{q}_i, \mathbf{q}_o)\}$ from $\mathcal{M}$ is stored into a qtDNN Replay Buffer. $N_{\text{qt}}$ tiny-batches, $\{\mathcal{B}_{\text{qt}}\}$, are sampled from $\mathcal{B}$ and then used to update qtDNN towards approximating PQC in $\mathcal{B}$. **c**, the updated qtDNN is then used in the surrogate model $Q_{\text{qt}}$ in this update step to facilitate the batched back-propagation of loss that are used to update the PreDNN and PostDNN with fixed qtDNN.

## 2.2 QTDNN: A TANGENTIAL PQC SURROGATE TRAINED ON-THE-FLY

We resolve this bottleneck by introducing the **qtDNN**, a quantum tangential space learning architecture that approximates the local behaviours of a QL with a classical DNN. The qtDNN is a small, differentiable surrogate network that locally mimics the PQC inside each mini-batch. It serves as a component of the hDQNN that substitutes the PQC whenever differentiability is needed.

The qtDNN is used during the training loop to provide a differentiable first-order approximation of PQC's behaviours, local to each mini-batch during the back-propagation. Meanwhile, the forward inference pass goes through the QLs. To make training the qtDNN more efficient, we add a replay buffer (qtDNN-buffer) to each QL that stores the input, $\mathbf{q}_i$, and sampled output, $\mathbf{q}_o$, of the quantum layer as a pair, $(\mathbf{q}_i, \mathbf{q}_o)$, from each mini-batch. This qtDNN-buffer is then used to train the qtDNN repeatedly with $N_{\text{qt}}$ tiny-batches randomly sampled from this qtDNN-buffer. This accelerates qtDNN's convergence to a moving approximation to its target quantum layer (cf. Haarnoja et al. (2018)).

The update epoch is split in two stages (Fig. 1b–c):

1. **Surrogate fit:** From the sampled mini-batch $\mathcal{B} = \{(\mathbf{q}_i, \mathbf{q}_o)\}$ we draw $N_{\text{qt}}$ tiny-batches to minimize a binary-cross-entropy between qtDNN($\mathbf{q}_i$) and the observed bit-strings $\mathbf{q}_o$.

2. **Network update:** We freeze the qtDNN, replace the PQC by it in a surrogate model $Q_{\text{qt}}$ that is otherwise identical to hDQNN, and run ordinary batched back-propagation to update the classical weights of the hDQNN.

Note that we refit the qtDNN at every actor update on the current mini-batch and freeze it only for the remainder of that update epoch; a justification and ablations are given in App. B.3. For complexity comparisons against parameter-shift and state-vector simulators, see App. A.1.2.

Because the qtDNN is itself a conventional MLP, gradients propagate without further quantum calls. The next subsection formalizes the approximation guarantee.

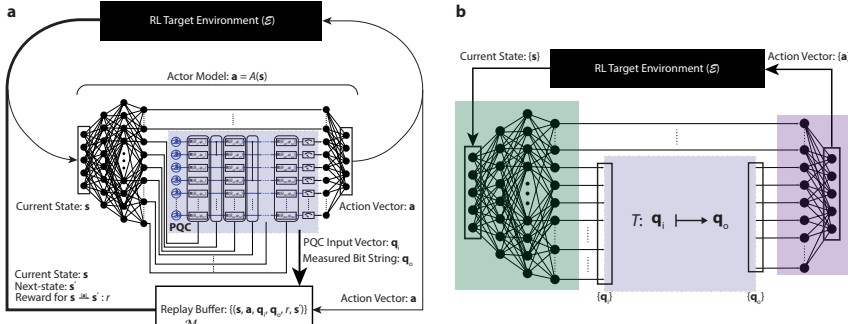

Figure 2: **a**, the typical exploration flow for the Agent to interact with the reinforcement learning objective's target environment, $\mathcal{E}$, and obtain experience data to be stored in the replay buffer, $\mathcal{M}$. **b**, the particular modular model architecture used for the Actor Model, $A$, that allows the intermediate mapping $T$ that maps its input vector $\mathbf{q}_i$ to $\mathbf{q}_o$ to be implemented differently without changing the rest of the model design. This allows comparing learning performance of different quantum and classical implementations fairly.

## 2.3 Local gradient fidelity of qtDNN

Intuitively, the PQC is a smooth function of its parameters over the small region explored by one mini-batch. A feed-forward MLP can capture its first-order behaviour. Theorem 1 states this precisely:

**Theorem 1** (Local gradient fidelity of qtDNN). *Let $Q : \mathbb{R}^d \to [0,1]^N$ be the deterministic map that returns the per-qubit $S$-shot marginal vector of the PQC, and suppose $Q \in C^1$ on an open set $\mathcal{U} \subset \mathbb{R}^d$. Fix a mini-batch $\mathcal{B} = \{x_i\}_{i=1}^{N_b} \subset \mathcal{U}$ lying in a closed ball of radius $r$ centered at $\bar{x} = \frac{1}{N_b}\sum_i x_i$. For every pair of tolerances $(\varepsilon_1, \varepsilon_2) > 0$ there exists a feed-forward MLP $Q_\theta : \mathbb{R}^d \to \mathbb{R}^N$ such that:*

$$\max_{x \in \mathcal{B}}\big\|Q_\theta(x) - Q(x)\big\| \leq \varepsilon_1, \qquad \max_{x \in \mathcal{B}}\big\|\nabla Q_\theta(x) - \nabla Q(x)\big\| \leq \varepsilon_2. \tag{1}$$

*Consequently, if the PQC is replaced by $Q_\theta$ inside the computation graph during back-propagation, the mini-batch gradient with respect to any upstream parameter vector $w$ is perturbed by at most an additive $O\big(\max(\varepsilon_1, \varepsilon_2)\big)$ term.*

The first inequality is simply the qtDNN learning to approximate the PQC's outputs correctly, while the second one shows that it learns to approximate the gradient of the PQC had it been differentiable i.e. approximate its effect during the back-propagation[2]. See App. B.1 for full proof details and App. D.1 for a pseudo-code constructing $Q$.

Our hardware-efficient ansatz uses only smooth rotations with fixed entanglers; the unitary and per-qubit expectations are analytic in the controls, hence $Q \in C^1$ on the region explored by a mini-batch (cf. App. B.2)

# 3 Quantum Reinforcement Learning

## 3.1 Problem set-up and motivation

We now instantiate the qtDNN surrogate inside a continuous–control agent (Fig. 2a). The target task is `Humanoid-v4` ($d_{\text{obs}} = 376, d_{\text{act}} = 17$), a long–horizon MuJoCo benchmark considered challenging for both classical and quantum methods ( Towers et al. (2024)). Classical baselines trained with identical compute on public HuggingFace repositories obtain final episode returns PPO $\approx 685$, SAC $\approx 5\,742$, TD3 $\approx 5\,306$ ( Zhao (2024),  Zhao (2023a),  Zhao (2023b)).

---

[2]We remark that Theorem 1 makes no assumption on the depth, connectivity, or gate layout of the underlying PQC; it guarantees local fidelity whenever the qtDNN fits the mini-batch outputs to within a target BCE loss. Empirical results suggest that this holds even for moderately deep and entangled circuits. We leave further tests on QAOA-style or chaotic ansätze as future work.

**Actor design.** The actor consists of a classical pre-network PreDNN, a middle block $T$, and a post-network PostDNN (Fig. 2b). To create a controlled ablation, we instantiate $T$ as the following separate blocks: a PQC (our hDQNN-TD3 model); a fully–connected layer (FC); a random bit generator (RBG); and an all-zero layer (0L).

*Note.* While $T$ is instantiated as the above for the ablation, the actor always contains the PreDNN and the PostDNN. For instance, in our experiments, the "FC model" has 5 layers: 2 layers from the PreDNN, 1 layer for $T$, and 2 layers from the PostDNN.

## 3.2 HDQNN-TD3 OVERVIEW

We start from Twin-Delayed DDPG (TD3) because it is competitive yet simple. As the Actor Model is what drives the actions, a more generalizable Actor Model will directly impact the performance. Therefore, the original TD3 critic remains classical, only the actor contains the PQC (or its surrogate). The qtDNN is trained inside each mini-batch to approximate the PQC locally, enabling standard batched back-propagation through the whole actor. We call the resulting model **hDQNN-TD3**. Fig. 2 sketches the interaction loop between the Agent and the classical environment $\mathcal{E}$.

For clarity we recap the four alternating phases that occur during training, the symbols follow the notation introduced in Subsec. 2.2. See App. D.2 for the pseudo-code of the training loop.

**1. Exploration.** At every environment step the current state $\mathbf{s} \in \mathcal{S}$ is encoded by the PreDNN. Its output is split into (i) classical bypass features $d_{\text{c-link}}$, and (ii) the control vector $\mathbf{q}_{\text{i}}$ that parameterises the PQC layer. Sampling the PQC yields the binary vector $\mathbf{q}_{\text{o}}$, which the PostDNN transforms, together with the bypass features, into the continuous action $\mathbf{a} \in \mathcal{A}$. Applying $\mathbf{a}$ in $\mathcal{E}$ returns the next state $\mathbf{s}'$, reward $r$, and a done flag. The transition tuple $(\mathbf{s}, \mathbf{a}, \mathbf{q}_{\text{i}}, \mathbf{q}_{\text{o}}, r, \mathbf{s}')$ is appended to the replay buffer $\mathcal{M}$.

**2. qtDNN update.** After $\mathcal{M}$ holds at least $N_b$ transitions we sample a mini-batch $\mathcal{B} \subset \mathcal{M}$. For every PQC in the Actor we collect the pairs $(\mathbf{q}_{\text{i}}, \mathbf{q}_{\text{o}})$ of the batch and train its qtDNN surrogate with $N_{\text{qt}}$ tiny-batches (from the same collection) using binary-cross-entropy. This freezes the qtDNNs for the remainder of the epoch.

**3. Critic update (TD learning).** With qtDNNs fixed, the surrogate Actor $A_{\text{qt}}$ provides target actions $\hat{\mathbf{a}} = A_{\text{qt}}(\mathbf{s}')$. The two critics minimize the TD-error $\epsilon_{\text{TD}} = \left\| r + \gamma\, C_t(\mathbf{s}', \hat{\mathbf{a}}) - C(\mathbf{s}, \mathbf{a}) \right\|^2$ as in TD3, using the loss averaged over $\mathcal{B}$.

**4. Actor update.** Every $N_{\text{A}}$ critic steps we update the classical parameters of the Actor by maximizing $C(\mathbf{s}, A_{\text{qt}}(\mathbf{s}))$ through standard back-propagation that now flows through the differentiable qtDNN, thereby avoiding expensive parameter-shift evaluations of the PQC. Target networks $C_t$ and $A_t$ are softly updated with rate $\tau$.

Moreover, Fig. 7 details the data flow inside one update epoch, making explicit which parameters are optimised at which stage.

## 3.3 QUANTUM–CLASSICAL INSTANTIATION DETAILS

The actor's middle block $T$ is a 10-qubit, 10-layer PQC. Each layer applies parameterised single-qubit rotations followed by at most one CZ entangler, and all controls come from the preceding PreDNN. Concretely:

- The input data is encoded in the controlling parameters $\{\theta_{ij}, \phi_{ij}, p_j, k_j\}$.

- $N = 10$ qubits, initialized in $|0\rangle^{\otimes N}$.

- $M = 10$ successive layers of gates.

- **Single-qubit stage.** For layer $j$ and qubit $i$ we apply $\mathbf{R}\big(\theta_{ij}, \phi_{ij}\big)$, where $\theta_{ij}, \phi_{ij} \in [0, 2\pi)$ are two outputs of PreDNN. Across the layer this uses $2N$ neurons.

- **Entangling stage.** The same layer chooses a control–target pair $(p_j, k_j) \in \{0, \dots, N-1\}^2$ by rounding two additional PreDNN outputs and applies a CZ gate if $p_j \neq k_j$ (identity otherwise).

- Hence PreDNN provides $(2N + 2)M = 220$ real controls per actor forward pass.

All remaining architectural details[3], including qtDNN width ($L_h = 2\,048$), critic topology, learning rates, and exploration noise, are listed in App. E.1.

**Noise regimes.** We consider (i) sampling noise from finite shots ($S \in \{100, 1000\}$), (ii) gate noise via bit/phase-flip channels with error rates $\sim 10^{-3}$, and (iii) exploration noise via Gaussian actions; qtDNN is trained directly on these noisy marginals; see App. F.5 for more details.

### 3.4 EXPERIMENTAL PROTOCOL

We train each agent for up to $7\,200$ episodes ($\approx 750\,000$ timesteps) across four seeds[4].

The qtDNN uses a single hidden layer of size $L_h = 2\,048$ and $N_{qt} = 32$ updates per epoch. Other parameters are exposed in Table 2. PPO/SAC/TD3 baselines follow the best Hugging Face RL models on official Gymnasium `Humanoid-v4` benchmark and consume the same number of steps to ensure a fair comparison.

**Ablation protocol.** For the ablation in §3.5.2 we keep the same total number of environment episodes and we note that there is an error rate $\sim 10^{-3}$ associated with each PQC gate. To isolate the contribution of the PQC, we compare the full hDQNN agent to control variants where the middle block is replaced by a fully connected layer (FC), a random bit generator (RBG), or a zero-output layer (0L). The FC variant matches the `qtDNN` in width and activation, ensuring that performance differences reflect the expressive power of the quantum layer rather than architectural mismatch. We consider $n_{qubits} \in \{5, 10\}$ and $n_{shots} \in \{100, 1\,000\}$. We omit the $(n_{qubits}, n_{shots}) = (5, 100)$ configuration due to unstable learning for some seeds (see Appendix A.1 for more details on limitations). A single-qubit setting would behave almost classically, whereas circuits with more than $\approx 20$ qubits can trigger peaks above 30 GB in GPU memory, leaving too little head-room on the 40 GB GPU (cf. §3, Google Quantum AI (2025)). Likewise, 10 shots are statistically insufficient and $10\,000$ shots yield negligible variance reduction relative to their cost.

### 3.5 RESULTS

#### 3.5.1 GENERAL PERFORMANCE

In the following part, we will observe the mean test return across training sessions. The test return is recorded by averaging episode rewards of 10 test episodes every $10\,000$ training steps (approximately 10 training episodes when the agent can constantly finish the whole $1\,000$ steps of an episode). The graphs show a 10-episode rolling average.

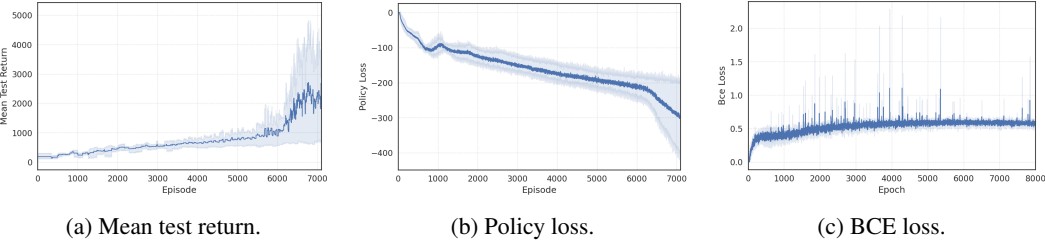

(a) Mean test return.  (b) Policy loss.  (c) BCE loss.

Figure 3: hDQNN-TD3 on `Humanoid-v4`; 4 seeds; 10-episode rolling average; 95% CI.

**Learning dynamics across seeds.** Fig. 3 aggregates the four independent training runs across $7\,200$ episodes and reveals three salient trends. First, the mean test return (Fig. 3a) rises almost monotonically after the initial warm-up, surpassing the $10^3$ mark around episode $3\,000$ and accelerating

---

[3]mini-batch size $N_b = 2^8$, actor update frequency $N_A = 2$.

[4]The horizon of $7,200$ episodes balances limited compute resources against the large number of ablation variants we evaluate. A single longer-horizon run is provided in §3.5.1.

sharply once the actor has experienced $5\,500$ episodes. The shaded $95\%$ confidence band widens in this late phase, indicating that one or two seeds surge ahead while the slower runs continue to improve, albeit more gradually. Second, the policy loss (Fig. 3b) mirrors this behaviour in reverse: it drops rapidly to $-100$ during the first $1\,000$ episodes, then decays almost linearly to $-200$ before steepening again in tandem with the return surge. The broader confidence interval after episode $5\,500$ signals larger gradient magnitudes and increased exploration of high-return regions. Finally, the concurrent monotonic gain in return and decline in loss across all seeds confirms that the qtDNN surrogate supplies gradients of sufficient fidelity throughout training, while the late-stage divergence illustrates the head-room the quantum-enhanced actor can exploit once it enters the high-performance regime. Finally, Fig. 3c demonstrates empirical evidence of Theorem 1 by plotting the BCE loss during the training:

- It begins with a warm-up phase: At epoch 0 the quantum-action vector $\mathbf{q}_{act}$ is almost zero, so the PQC outputs a highly imbalanced bit-string vector that the surrogate predicts with near-perfect accuracy. As exploration noise broadens the input distribution over the first $\sim 150$ epochs, BCE rises to $\approx 0.45$.

- Then, it reaches a stable plateau: From epoch 200 onward the loss settles in the 0.55-0.65 band and remains there for $> 99\%$ of the $8\,000$-epoch run. This is at least 13 % below the random-guess entropy of 0.693, i.e. each qubit is still predicted with $\sim 65$ % accuracy, this taking into account that the PQC is noisy.

- Throughout the training we notice transient spikes. Those isolated spikes occur when a mini-batch is label-skewed; they vanish within one update and do not accumulate.

This shows that the BCE loss is stable throughout the training. This suggests that the surrogate remained within the $\varepsilon$–ball required by Theorem 1 throughout training, supporting the assumption that gradient perturbations stayed within tolerance (see App. B for proof details and assumptions).

**Best-seed comparison with benchmark.** Fig. 4 contrasts the best seed of each actor variant over a longer $14\,000$-episode horizon[5]. The hDQNN-TD3 actor (solid blue) climbs steeply after episode $5\,500$ and saturates near a $6\,011 \pm 83$ return, surpassing the best seeds of classical TD3 ($5\,306$), SAC ($5\,742$) and PPO ($685$). Two additional aspects are noteworthy:

- *Sample-efficiency.* hDQNN-TD3 reaches the $3\,000$-return mark about 800 episodes earlier than its fully connected (FC) alternative, suggesting that the quantum layer helps discover high-value policies sooner.

- *Stability of convergence.* The hDQNN-TD3 curve flattens into a narrow band after episode $11\,000$, whereas the FC and RBG variants continue to exhibit larger oscillations, this indicates that the qtDNN surrogate supplies gradients of sufficient fidelity to stabilize late-stage policy refinement.

Overall, the figure supports our claim that, in the best case, a single well-tuned hDQNN-TD3 seed can match the ceiling performance of the strongest classical baselines while requiring fewer interaction steps to get there.

**Other environments.** We tested the hDQNN-TD3 architecture on the `Hopper-v4` environment (lower-dimensional, single-leg locomotion) across five seeds. Figure 5a shows that our model is able to reach a mean return on par with TD3's performance on the former `Hopper-v1` benchmark, namely stabilizing around a 3400 score after around 5000 epochs. Meanwhile, Figures 5b and 5c show the stability of the learning with a monotonously decreasing policy loss and a stable BCE loss throughout the training process.

### 3.5.2 ABLATION STUDY

Table 1 contrasts five actor variants at two training checkpoints[6]. The upper block (Episode $4\,000$) gives a mid-training snapshot. Already at this stage the 10-qubit, $1\,000$-shot PQC attains a mean

---

[5]For clarity the PPO, SAC and TD3 reference values from best Hugging Face RL models (Zhao (2024; 2023a;b)) on Gymnasium `Humanoid-v4` are shown as horizontal dashed lines.

[6]Since the test return is noisy, we used the test return's 10-episode rolling average across episodes to smoothen it and avoid recording misleading noise at the checkpoint.

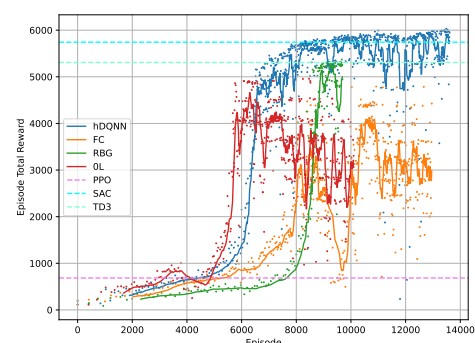

Figure 4: Best seed return; equal step budget; horizontal lines are public PPO/SAC/TD3 references.

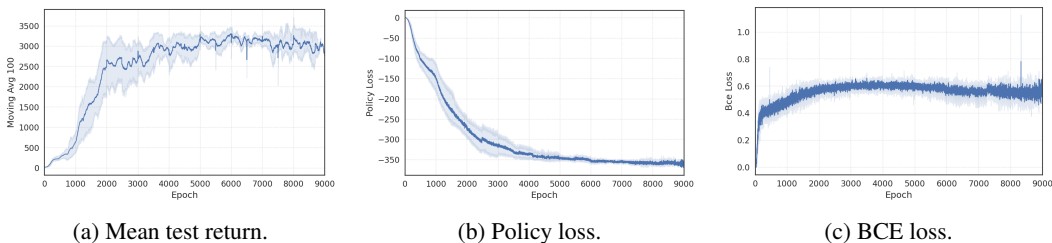

(a) Mean test return.  (b) Policy loss.  (c) BCE loss.

Figure 5: hDQNN-TD3 on `Hopper-v4`; 4 seeds; 10-episode rolling average; 95% CI.

return of $678.7 \pm 179.3$, and its best seed reaches $926.7$, beating the FC by $+296$ (mean-to-mean) and the RBG by $+322$.

The lower block (Episode $7\,200$) captures late-training behaviour. Here, the 5-qubit, $1\,000$-shot PQC leads with a mean return of $1\,132.6 \pm 1\,411.4$ and a best seed of $3\,211.6$, maintaining a $+512$ margin over FC and $+706$ over RBG. The larger 10-qubit variant still improves on FC by $+149$ despite its higher circuit cost.

Replacing the PQC with either an RBG or a deterministic FC layer consistently lowers final performance by hundreds of return points, confirming that the quantum layer contributes non-trivial modeling capacity rather than acting as a mere regulariser.

Across seeds, PQC outperforms the strongest FC baseline on average by $+296$ at episode $4,000$ and by $+148$ at episode $7,200$ (Table 1), confirming that the advantage is not seed-specific.

This consistent increase of performance between the FC and the PQC layer shows that the qtDNN is able to approximate the QL during back-propagation: the model is able to make use of the more complex representation of the QL during forward-prop and achieve a higher return, all while keeping stable gradients during back-propagation. Although not a direct experimental proof, this observation goes in the way of validating the local-tangent result of Theorem 1, i.e. $\|\nabla Q_\theta - \nabla Q\| \leq \varepsilon$ across the training. Two theoretical lenses that explain the advantage over an equally wide FC layer are summarised in App. G.

Although qtDNN is trained on each mini-batch independently, results in Table 1 show consistent performance across seeds with different replay distributions, suggesting robustness to moderate batch variability. Nonetheless, investigating further how buffer diversity affects surrogate generalisation is an important research direction, which we leave for future work.

### 3.5.3 QUANTUM CALLS

The PQC calls column from Table 1 shows how many quantum calls have been performed during the training. This number, $K$, is the same as the number of total steps the agent took during the training to explore its environment and populate the replay buffer, $\mathcal{M}$, for updating classical networks in

Table 1: Ablation study on `Humanoid-v4`. Test returns are averaged over 4 seeds.

| Episode | Layer | Shots | Qubits | Mean return | Best return | Time[min] | PQC calls |
|---------|-------|-------|--------|-------------|-------------|-----------|-----------|
| 4 000 | RBG | 0 | 0 | $356.5 \pm 219.1$ | 545.3 | 39.8 | 0 |
| 4 000 | FC | 0 | 0 | $382.7 \pm 167.5$ | 603.6 | 27.4 | 0 |
| 4 000 | PQC | 100 | 10 | $470.8 \pm 57.1$ | 534.5 | 291.2 | 302 738 |
| 4 000 | PQC | 1000 | 5 | $521.3 \pm 128.0$ | 650.8 | 227.3 | 302 948 |
| 4 000 | PQC | 1000 | 10 | $678.7 \pm 179.3$ | 926.7 | 307.3 | 359 952 |
| 7 200 | RBG | 0 | 0 | $426.8 \pm 114.5$ | 545.0 | 105.9 | 0 |
| 7 200 | FC | 0 | 0 | $620.8 \pm 393.8$ | 1 207.8 | 89.8 | 0 |
| 7 200 | PQC | 100 | 10 | $517.8 \pm 95.0$ | 659.2 | 712.2 | 694 104 |
| 7 200 | PQC | 1000 | 5 | $1132.6 \pm 1411.4$ | 3 211.6 | 737.9 | 769 867 |
| 7 200 | PQC | 1000 | 10 | $768.9 \pm 387.4$ | 1 277.4 | 700.9 | 761 498 |

off-policy training process. Each forward inference through the Actor Model makes one PQC call. Each PQC call runs the circuit for $S$ shots and returns the per-qubit empirical marginal vector $\mathbf{q}_o$ computed from those $S$ bit-strings. Thus, training with qtDNN performs exactly $K$ PQC calls and $S \times K$ shot executions on an expensive physical QPU. Meanwhile, alternative quantum hardware compatible gradient methods like parameter shifts require $\sim I \times O \times N_b \times S \times K$ additional physical QPU runs over the training process for our QL with $I = 220$ dimensional input, $O = 10$ dimensional output, and a mini-batch size of $N_b = 2^8$ in our cases.

**Numerical efficiency.** Training with qtDNN performs exactly $K$ PQC calls and $S{\cdot}K$ shots (forward only), whereas parameter–shift would require $\tilde{O}(I \cdot O \cdot S \cdot N_b \cdot K)$ additional QPU runs. In our $N{=}10$, $M{=}10$ setting ($I{=}220$, $O{=}10$, $N_b{=}2^8$), even a light budget with $S{=}10$ implies $IOSN_b \approx 5.63 \times 10^6$ shots per update, i.e., $\approx 3.38$s at $\sim 0.6\,\mu s$ per shot; over $7.5 \times 10^5$ updates ($\sim$7.2k episodes) this is $\gtrsim 2.5 \times 10^6$s ($\sim$29 days), while our qtDNN run finished in $\sim$700 minutes (App. A.1.2).

## 4 DISCUSSION

In this work, we proposed a novel way to efficiently train hDQNN compatible with NISQ quantum hardware, allowing for high-throughput mini-batch training using a surrogate classical neural network (qtDNN) to locally approximate and estimate gradients of the embedded quantum circuit for each mini-batch parameter update in off-policy reinforcement learning process. This method only requires expensive PQC calls in inference steps and therefore minimizes the number of expensive PQC calls in training significantly while still allows for efficient parallel mini-batch back-propagation to update classical weights. We note that surrogate-assisted training reduces quantum-specific wall-clock by more than an order of magnitude compared with parameter-shift; see App. A.1.2 for a detailed cost model and lighter-budget variants. We further constructed hDQNN-TD3 model to learn complex, high-dimensional, and continuous-variable reinforcement learning tasks like `Humanoid-v4` successfully with a final reward score comparable to that of models with widely used SOTA architectures except the recently proposed MEow from Chao et al. (2024). We further carried out ablation study by swapping the PQC to alternative classical layers and observed consistently better performance with hDQNN-TD3 against all comparable classical alternatives.

Therefore, training hDQNN with qtDNN and inference with trained hDQNN on (noisy) quantum hardware offer a new route in reinforcement learning and AI research for addressing complicated real-world problems that are always stochastic, highly correlated, and even challenging to simulate classically. Further research in this direction could lead to novel models such as hDQNN-LLM where classical layers responsible for high-dimensional encoding and sampling in LLMs, such as DeepSeek-AI et al. (2025a), are replaced with QLs surrounded by classical DNNs to learn the strongly correlated distributions in natural language as hinted by Lo et al. (2023). Moreover, the hDQNN-TD3 could be further developed to enhance models like NVIDIA (2025) in controlling advanced real-world robots.

**Ethics Statement.**    This work does not involve human subjects or personally identifiable data. Potential dual–use considerations arise from quantum computing research in general, but the presented method is limited to reinforcement–learning benchmarks and carries no direct security or safety risks. We note that training with large quantum simulations consumes substantial compute, and future work should continue to assess energy efficiency.

**Reproducibility Statement.**    We will share an open-source repository with the full code and configuration files upon acceptance of the paper. All experiments were run on NVIDIA A100–SXM4 GPUs (40 GB HBM2e). Software stack: Python 3.12.3, PyTorch 2.6.0 (CUDA 12.4), Gymnasium 1.1.1 with MuJoCo 3.3.2, and CUDA–Q 0.9.1. Protocol (extended in appendix): 4 random seeds, 7,200 episodes ($\approx 7.5 \times 10^5$ steps), mini–batch size $N_b{=}28$, actor update every $N_A{=}2$ critic steps, and $N_{\mathrm{qt}}{=}32$ surrogate updates per epoch. All hyperparameters and ablation settings are documented in App. E.1, E.2. Figures and tables include mean and $95\%$ confidence intervals across seeds.

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

# A  APPENDIX

## A.1  LIMITATIONS

### A.1.1  TRAINING HDQNN WITH QTDNN SURROGATE

The assumption that a quantum tangential deep neural network (qtDNN) can be trained as a good approximation to a given QL can only be reasonable for each mini-batch of training data (see Fig. 7) in off-policy reinforcement learning frameworks. This is because the sampled distribution of a general PQC is exponentially hard to simulate with classical computing resources Arute et al. (2019); Acharya et al. (2025).

Moreover, during the experiment, we implemented qtDNN as a 3-layer MLP and observed that training is most efficient when the hidden layer spans a linear space $\mathcal{L}_{\mathrm{h}} = \mathbb{R}^{2^{N+1}}$ with $2^{N+1}$ neurons for a $N$-qubit wide PQC. As $N$-qubit spans a Hilbert space $\mathcal{H}_{Nq} \cong \mathbb{C}^{2^N} \cong \mathbb{R}^{2^{N+1}} \cong \mathcal{L}_{\mathrm{h}}$, the observation indicates that physics-inspired accurate PQC state representation by the hidden layer can guide the design of optimal qtDNN. Therefore, it is reasonable to expect that the number of neurons required for 3-layer MLP qtDNN scales according to $2^{N+1}$ with $N$ qubits in the PQC it approximates locally. We also note that the optimal 3-layer MLP design of qtDNN seems to be unexpectedly independent of PQC depth, $M$. This exponential resource scaling would limit the 3-layer MLP implementation of qtDNN to relatively narrow (in qubit count) PQC. The limitation on PQC width could further limit the ability for hDQNN-TD3 to learn complex AI/ML tasks involving a vast number of correlated degrees of freedoms and hinder its applications in the Large Language Models (LLMs) DeepSeek-AI et al. (2025a;b) and advanced robotic controls in unpredictable real-world interactive conditions as in NVIDIA (2025). Future development of novel qtDNN architecture might significantly improve the scaling efficiency by tailoring the qtDNN design to respect the structure of a given PQC and bound its resource requirements with a function of $N_{\mathrm{b}}$. For example, tensor network methods could approximate a PQC with a given structure efficiently Liu et al. (2024); Huang et al. (2021a). However, even with the potentially large resource requirements scaling exponentially with the number of qubits involved in sampling, we could argue that this limitation is only present in training and the inference data flow of a trained hDQNN is potentially exponentially more efficient as forward pass through the PQC could be carried out on physical QPUs.

To counter the width scaling of the plain MLP we used in the qtDNN, techniques such as tensor-network surrogates, pruning, or symmetry-aware PQC compression could be used to significantly reduce the parameter footprint without affecting training fidelity. We view these optimisations as complementary to the main goal of demonstrating the feasibility of surrogate-assisted hybrid quantum training and leave their application as future work.

For the inference, physical QPUs could carry out PQC operations directly on all qubits without the requirement of exponentially scaling physical computing resources Acharya et al. (2025). Moreover, the superconducting QPU's 10s ns level quantum entangling gate times Gao et al. (2025) are preferred for co-processing with classical neural networks because the gap in clock rates is much smaller as compared to trapped ion and neutral atom platforms' average entangling gate times around 0.5 ms. Here, we note that even for SOTA high-connectivity superconducting quantum processors, each PQC call with $S = 1,000$ will take about $t_{\mathrm{qc}} \sim M \times 50$ ns $+ 100$ ns $\sim 0.6$ $\mu$s (assuming a single-qubit and a two-qubit gates in sequence take 50 ns and measurement take 100 ns), leading to a per PQC Call to backend with $S = 1,000$ shots consuming $S \times t_{\mathrm{qc}} \sim 0.6$ ms in inference step. Therefore, having fast gates is practically useful for quantum reinforcement learning tasks.

In terms of applying this hDQNN-TD3 method to a variety of different RL environments, only `Humanoid-v4` and the simple `Hopper-v4` were attempted due to time and resource limitations. Even though they represent two extremes of complexities in Gymnasium continuous variable reinforcement learning benchmarks and achieved good results, the long PQC simulation time became a bottleneck making it challenging to extend the experimentation to other environments within the short time available to prepare the paper. However, it is straightforward to apply hDQNN-TD3 for enhancing reinforcement learning in other useful scenarios involving both continuous and discrete observation and action spaces. Integration with the trailblazing AI models' training and inference pipelines DeepSeek-AI et al. (2025a) could be among the topics in future researches.

In the end, hDQNN requires PQC to be executed on a physical or simulated QPU. Real-world QPUs in the near-term have noises (errors) in quantum operations they carry out. Therefore, in this work, we tested the noise tolerance of hDQNN-TD3 method in learning `Humanoid-v4` on top of the Gaussian action noise. The hDQNN-TD3 model having PQC bit-flip and phase-flip error rates of $10^{-3}$ perform on par and even potentially better than classical SOTA models in terms of rewards they achieved given a similar compute. However, a larger $10^{-2}$ error rate in physical quantum gates would lead to a final reward lower than SOTA models. On the other hand, we further noticed hDQNN-TD3 model demonstrated faster learning with elevated error rates in quantum gates and attribute this to quantum gate noises (errors) having the same effect of larger exploration noises in classical SOTA models. We note that hyperparameters are kept the same between comparable alternative quantum and classical models presented in the main text. We further note that, as we took the most probable bit-string from sampling PQC over $S$ identical shots as the output $\mathbf{q}_\mathrm{o}$ of a QL in hDQNN for a given input, $\mathbf{q}_\mathrm{i}$, reducing the number of shots $S$, in each sampling for estimating $\mathbf{q}_\mathrm{o}$ would increase the noises in $\mathbf{q}_\mathrm{o}$. This increased sampling noise impacts the results in a similar way as errors in quantum operations.

### A.1.2 The inefficiency of training hDQNN without qtDNN surrogate

Because $I \times O$ different circuits are required to estimate all the gradients of a QL with $I$ input dimensions and $O$ output dimensions and each circuit needs a number, $S$, of shots to have the statistics for estimating the observables used to calculate gradients. Moreover, learning for complex problems with a high sample data efficiency would require training data to be sampled from past experiences in mini-batches of size $N_\mathrm{b}$ for each DNN update with batch optimisation as in Haarnoja et al. (2018). Therefore, existing QML methods that directly estimate gradients of the QL are faced with a prohibitive practical training overhead proportional to $I \times O \times S \times N_\mathrm{b}$ for running on real QPU that can only run one shot at a time. For example, in the case of using a fast superconducting Josephson junction quantum processor that has gate time as short as 20 ns and measurement time in the range of 100 ns as seen in Acharya et al. (2025); Gao et al. (2025), a 10-layer 10-qubit PQC (QL) would need $I = 220$ and $O = 10$ with $N_\mathrm{b} = 256$ and $S = 1000$ in our example for learning complex `Humanoid-v4` environment. In this case, each update to model DNN parameters would require about 281.6 s. Considering that typical classical RL training even for simple tasks would likely require $\sim 10^6$ update steps similar to Haarnoja et al. (2018), this would translate to more than 10 years of training.

**A lighter but still impractical parameter-shift budget.** Consider an optimistic configuration with $S = 10$ shots per circuit, $N_b = 256$, $I = 220$ inputs and $O = 10$ outputs for the $N{=}10, M{=}10$ PQC, and single-/two-qubit/measurement latencies of $(50\,\mathrm{ns}, 50\,\mathrm{ns}, 100\,\mathrm{ns})$. Each actor update requires $I \times O \times S \times N_b \approx 5.63 \times 10^6$ shots, i.e. $\approx 3.38\,\mathrm{s}$ at $\sim 0.6\,\mu\mathrm{s}$ per shot. Over $\sim 7.5 \times 10^5$ updates (7 200 episodes), this exceeds $2.5 \times 10^6\,\mathrm{s}$ ($\approx 29$ days) before queueing/calibration overhead. By contrast, the qtDNN run in Table 2 completes in $\sim 700$ minutes.

### A.1.3 qtDNN versus classical simulators.

A noiseless state-vector simulator stores $2^N$ amplitudes and updates them gate-by-gate, with time/memory complexity $O(2^N L)$. Even modest circuits thus exceed the footprint required for parallel mini-batches. More importantly, gradients from an ideal simulator would ignore read-out error, decoherence, and cross-talk present at inference on NISQ QPUs. In contrast, qtDNN is fitted on the noisy measurements $\mathbf{q}_\mathrm{o}$ at every step, so the gradients used to train the actor are aligned with the behaviour that the policy will experience at deployment.

### A.1.4 Complexity of qtDNN updates

One surrogate fit touches $W$ weights, hence costs $O(W)$ FLOPs per tiny-batch. In our 10-qubit setup with PostDNN bypass of $d_\mathrm{c\text{-}link} = 10$ and a single hidden layer $L_h = 2048$, the surrogate has $W \approx 2048\,(2N{+}2)M + 2048\cdot 10 + 2048 \cdot 10 \approx 4.7 \times 10^5$ parameters ($\approx 1.8\,\mathrm{MB}$ in FP32). By contrast, an exact state-vector simulator stores $2^N$ complex amplitudes and updates them per gate, $O(2^N L)$, while a parameter-shift estimator incurs $\Theta(I)$ additional circuit evaluations per input dimension, returning to $O(I\,2^N L)$ time and memory. This explains the 2–3 orders of magnitude gap

between our qtDNN updates and simulator-based gradients at the $N{=}10$, $L{=}10$ scale reported in Table 3.

## B  LOCAL GRADIENT FIDELITY

### B.1  PROOF OF THEOREM 1 ON THE LOCAL GRADIENT FIDELITY OF QTDNN

**Summary of proof.** Because the mini-batch $\mathcal{B}$ lies inside a small radius-$r$ ball, $Q$ is well-approximated by its first-order Taylor expansion around the centre $\bar{x}$. Thus, we (i) approximate the first-order Taylor polynomial $P(x):=Q(\bar{x})+J_Q(\bar{x})(x-\bar{x})$ over $\mathcal{B}$ by an MLP $Q_\theta$, and (ii) use Lipschitz bounds on the second derivative of $Q$ to control the Taylor remainder and its gradient. Replacing the PQC by that MLP therefore changes (i) forward activations by at most $\varepsilon_1$ and (ii) their Jacobian by at most $\varepsilon_2$, which in turn perturbs the gradient flowing into earlier layers by the same order of magnitude. See below for the full proof.

*Proof.* Let $\mathcal{B} \subset \mathbb{R}^d$ be the mini-batch, contained in the closed ball $\overline{B}_r(\bar{x})$. Because $Q$ is $C^1$ on an open neighbourhood of $\overline{B}_r(\bar{x})$, its Jacobian $J_Q$ is Lipschitz–continuous: we can fix a constant $L > 0$ such that $\|J_Q(x) - J_Q(y)\| \leq L\|x - y\|$ for all $x, y \in \overline{B}_r(\bar{x})$. Set $M = \|\nabla^2 Q(\bar{x})\|$ and $\delta = \max(\varepsilon_1, \varepsilon_2) - \frac{1}{2}Lr^2 > 0$.

**Step 1. Approximating the affine Taylor part.** Write $h = x - \bar{x}$. First-order Taylor expansion gives $Q(x) = Q(\bar{x}) + J_Q(\bar{x})h + \rho(x)$, where the remainder satisfies the integral form

$$\rho(x) = \int_0^1 \big[J_Q(\bar{x} + th) - J_Q(\bar{x})\big]h\,dt. \tag{1}$$

Define the affine map $P(x) := Q(\bar{x}) + J_Q(\bar{x})h$. The universal-approximation theorem ensures that there exists an MLP $Q_\theta$ such that

$$\max_{x \in \mathcal{B}} \|Q_\theta(x) - P(x)\| \leq \delta, \qquad \max_{x \in \mathcal{B}} \|\nabla Q_\theta(x) - J_Q(\bar{x})\| \leq \delta. \tag{2}$$

**Step 2. Bounding the remainder $\rho$ and its gradient.** Because $J_Q$ is $L$-Lipschitz and $\|h\| \leq r$, from equation 1 we obtain

$$\|\rho(x)\| \leq \int_0^1 L\,t\,\|h\|^2\,dt = \frac{1}{2}L\|h\|^2 \leq \frac{1}{2}Lr^2. \tag{3}$$

Differentiating equation 1 with respect to $x$ yields, using $\nabla_x h = I$,

$$\nabla\rho(x) = \int_0^1 t\nabla^2 Q(\bar{x} + th)\,h\,dt.$$

Hence

$$\|\nabla\rho(x)\| \leq \int_0^1 t\|\nabla^2 Q(\bar{x} + th)\|\|h\|dt \leq \frac{1}{2}\big(M + Lr\big)r. \tag{4}$$

**Step 3. Uniform error bounds on $\mathcal{B}$.** For any $x \in \mathcal{B}$, combine equation 2 with equation 3:

$$\|Q_\theta(x) - Q(x)\| \leq \underbrace{\|Q_\theta(x) - P(x)\|}_{\leq \delta} + \underbrace{\|\rho(x)\|}_{\leq \frac{1}{2}Lr^2} \leq \delta + \frac{1}{2}Lr^2 = \varepsilon_1.$$

Likewise, using equation 2 and equation 4,

$$\|\nabla Q_\theta(x) - \nabla Q(x)\| \leq \underbrace{\|\nabla Q_\theta(x) - J_Q(\bar{x})\|}_{\leq \delta} + \underbrace{\|\nabla\rho(x)\|}_{\leq \frac{1}{2}(M+Lr)r} \leq \delta + \frac{1}{2}(M + Lr)r = \varepsilon_2.$$

**Step 4. Impact on back-propagation.** Inside the computation graph, replacing $Q$ by $Q_\theta$ changes the per-sample loss by at most $\varepsilon_1$, and perturbs every mini-batch gradient entering the upstream network by at most $\varepsilon_2$. Thus all parameter updates incur an additive $O\big(\max(\varepsilon_1, \varepsilon_2)\big)$ error, completing the proof.

$\square$

### B.2 Smoothness of PQC expectations.

In our hardware-efficient ansatz all tunable gates are smooth rotations $R_\alpha(\theta) = \exp(-i\theta\,\sigma_\alpha/2)$ and entanglers are fixed; composition and the exponential map are analytic. Therefore the unitary $U(\boldsymbol{\theta})$ and the expectations $\langle Z_k\rangle(\boldsymbol{\theta}) = \langle 0|U^\dagger(\boldsymbol{\theta})Z_k U(\boldsymbol{\theta})|0\rangle$ are $C^\infty$ in $\boldsymbol{\theta}$, which justifies the hypothesis $Q \in C^1$ used in Theorem 1. If one inserts non-smooth operations (e.g., hard thresholds or topology-switching branches), $Q$ may only be piecewise differentiable; in that regime the tight mini-batch gradient fidelity does not apply verbatim and one must resort to sub-gradient or sampling-based estimators with typically higher variance.

### B.3 Refitting vs freezing the surrogate.

After each actor update step the policy parameters change, shifting the distribution of PQC inputs seen at the next update. Holding the surrogate fixed across epochs would increase its BCE on the new batch, risking violation of the $\varepsilon$-ball required by Theorem 1. Because a qtDNN fit accounts for only $\sim 2\%$ of wall-clock on our A100 runs (App. G.3), we refit *every* actor update and freeze the surrogate only for the remainder of that update epoch.

## C  Mathematical formalisation of the hDQNN-TD3's components

The parameterised quantum circuit (PQC) in the hDQNN-TD3 is a variational circuit designed for NISQ devices, utilizing $N$ qubits to act as a quantum hidden layer within the actor network, taking its inputs from the DNN's previous layer's extracted features. It consists of $M$ layers, each comprising single-qubit rotation gates and entangling CZ gates.

Formally, the PQC transforms an input vector $\mathbf{q_i}$ into a probability distribution over bit-strings $\mathbf{q_o}$ by the following:

$$|\psi(\mathbf{q_i})\rangle = U(\mathbf{q_i})|0\rangle^{\otimes N} \tag{2}$$

Where $U$ is a sequence of parameterised quantum gates acting on the $N$ qubits. (initially in the $|0\rangle^{\otimes N}$ states):

$$U(\mathbf{q_i}) = \prod_{l=1}^{M}\left[\prod_{i,j\in\{1,\ldots,N\}} \mathrm{R}(\theta_{i,l}, \phi_{i,l})\mathrm{O}_{\{i,j,l\}}\right] \tag{3}$$

Here, $\mathrm{R}(\theta_{i,l}, \phi_{i,l})$ is a single-qubit rotation gate around the $X$-axis and $Z$-axis on qubit $i$ in layer $l$ by $\theta_{i,l}$ and $\phi_{i,l}$ degrees respectively.

Additionally, $\mathrm{O}_{\{i,j,l\}}$ represents an entangling operator acting on the qubits $i$ and $j$ in the $l$ layer when $i \neq j$. When $i = j$ we define $\mathrm{O}_{\{i,j,l\}} = \mathbb{I}$ as an identity operator. For instance, we can pose $\mathrm{O}_{\{i,j,l\}} = \mathrm{CNOT}_{i,j,l}$ which is a controlled-NOT gate between qubits $i$ and $j$. In the `hDQNN-TD3`, $\mathrm{O}_{\{i,j,l\}} = \mathrm{CZ}_{i,j,l}$.

Altogether, these gates constitute the unitary $U(\mathbf{q_i})$, which acts on the $N$-qubit initial state $|0\rangle^{\otimes N}$ (or a suitably encoded feature state) to produce the output state $|\psi(\mathbf{q_i})\rangle$.

When measuring the qubits in the computational basis, one obtains a bit-string $\mathbf{q_o} \in \{0,1\}^N$, with probability:

$$P(\mathbf{q_o} \mid \mathbf{q_i}) = \big|\langle \mathbf{q_o} \mid \psi(\mathbf{q_i})\rangle\big|^2 \tag{4}$$

## C.1 Equations for the **qtDNN** surrogate

**Problem statement.** During each *forward* pass the real PQC (or its CUDA–Q simulation) is executed to generate a bit-string $\mathbf{q_o} \in \{0,1\}^N$; the qtDNN is invoked *only in the backward pass* as a differentiable proxy that supplies $\nabla_\theta \hat{\mathbf{p}}$ without additional quantum evaluations. To this end we draw mini-batches of pairs $(\mathbf{q_i}, \mathbf{q_o})$ where $\mathbf{q_i} \in \mathbb{R}^{2NM} \bigotimes \mathbb{N}^{2M}$ are the rotation angles and pairs of qubit indices fed to the PQC and $\mathbf{q_o} \in \{0,1\}^N$ is a bit-string obtained from measurements over multiple shots of executing the same quantum circuit. In hDQNN-TD3, the bit-string is taken as the most probable string among the bit-strings obtained from multiple shots. The qtDNN learns a mapping

$$f_{\text{qtDNN}} : \mathbf{q_i}, \omega \longmapsto \hat{\mathbf{p}} = (\hat{p}_{\omega,1}, \ldots, \hat{p}_{\omega,N}), \qquad \hat{p}_{\omega,k} \in (0,1), \tag{5}$$

where $\omega$ denotes the qtDNN parameters and $\hat{p}_{\omega,k}$ is the surrogate's estimate of the marginal probability that qubit $k$ is measured in state $|1\rangle$.

**Training objective.** Ideally one would minimise the cross-entropy with respect to the *joint* distribution produced by the PQC, $P_{\text{PQC}}(\mathbf{q_o} \mid \boldsymbol{\theta})$, namely $\mathcal{L} = -\log P_{\text{PQC}}(\mathbf{q_o} \mid \boldsymbol{\theta})$. However estimating that joint requires $\Omega(2^N)$ shots per input and is infeasible even for the moderate $N = 10$ qubits used in our study. Following prior work on quantum-model surrogates, we therefore adopt the factorised *binary cross-entropy* (BCE)

$$\mathcal{L}_{\text{qtDNN}}(\omega) = -\sum_{k=1}^{N} \Big[ \mathbf{q_o}_k \log \hat{p}_{\omega,k} + (1 - \mathbf{q_o}_k) \log(1 - \hat{p}_{\omega,k}) \Big], \tag{6}$$

$$= \mathbb{E}_{(\boldsymbol{\theta}, \mathbf{q_o}) \sim \mathcal{M}} \big[ \text{BCE}(\mathbf{q_o}, f_{\text{qtDNN}}(\mathbf{q_i}; \omega)) \big], \tag{7}$$

where $\mathcal{M}$ is the mini-batch of replay buffer of observed $(\mathbf{q_i}, \mathbf{q_o})$ pairs.

**Independence assumption.** Equation equation 6 treats qubits as independent Bernoulli variables, whereas the true PQC may entangle them. We demonstrated theoretically that this approximation is adequate for gradient propagation. Moreover, the stability of the learning process shows that the surrogate successfully reproduces the expectation values of the PQC. Future work could replace the factorised model with an autoregressive density estimator to capture higher-order correlations without the exponential shot overhead.

## C.2 Forward pass of the hDQNN (inference, no RL)

The hybrid actor is evaluated in three deterministic stages; no reinforcement–learning logic or qtDNN surrogate is involved in this pure inference path.

**1. PreDNN $\longrightarrow$ rotation angles.** Given an observation $\mathbf{x} \in \mathbb{R}^n$, the `PreDNN` (a feed-forward MLP) produces a latent vector $\mathbf{z}_{\text{pre}} \in \mathbb{R}^{(2N+2)M+d_{\text{c-link}}}$, where the last $d_{\text{c-link}}$ elements are passed to `PostDNN` directly. The rest of the elements are rescaled to $[-\pi, \pi]^{\otimes 2NM}$ or rounded to integers $\mathbb{N}^{2M}$ as follows. The rescaled part is split into the pair of rotation angles for every qubit in every layer of the PQC,

$$\boldsymbol{\theta} = \mathbf{z}_{\text{pre}}^{1:N \times M}, \qquad \boldsymbol{\phi} = \mathbf{z}_{\text{pre}}^{N \times M+1:2N \times M}, \qquad \boldsymbol{k} = \mathbf{z}_{\text{pre}}^{2N \times M+1:(2N+2)M},$$

$$\mathbf{q_i} = (\boldsymbol{\theta}, \boldsymbol{\phi}, \boldsymbol{k}) \in [-\pi, \pi]^{2N \times M} \otimes \mathbb{N}^{2M}. \tag{8}$$

**2. PQC $\longrightarrow$ expectation values.** The rotation-encoded angles drive the parameterised quantum circuit $U(\mathbf{q_i})$, producing the state $|\psi(\mathbf{q_i})\rangle$. Instead of sampling a bit-string, the implementation measures the *Pauli-Z* expectation of each qubit,

$$z_k^{(q)} = \langle \psi(\mathbf{q_i}) | Z_k | \psi(\mathbf{q_i}) \rangle, \qquad k = 1, \ldots, N, \tag{9}$$

yielding the vector $\mathbf{z}_q = (z_1^{(q)}, \ldots, z_N^{(q)}) \in {-1, 1}^N$. A single bit-string sample is drawn only for logging purposes.

**3. PostDNN $\longrightarrow$ action prediction.** An optional "skip" connection concatenates the last $d_{\text{c-link}} \leq N$ entries of $\mathbf{z}_{\text{pre}}$, $\mathbf{z}_{\text{c-link}}$, to the quantum layer output bit-string:

$$\tilde{\mathbf{z}} = \begin{bmatrix} \mathbf{z}_{\text{q}} \,\|\, \mathbf{z}_{-\text{link}} \end{bmatrix}.$$

The resulting feature vector $\tilde{\mathbf{z}}$ is fed to `PostDNN`, producing the final continuous action

$$\mathbf{y}_{\text{pred}} = f_{\text{PostDNN}}(\tilde{\mathbf{z}}) \in \mathbb{R}^{n_a}. \tag{10}$$

**Summary.** Putting the three stages together, the deterministic forward map of the actor is

$$\boxed{\mathbf{x} \xrightarrow{f_{\text{PreDNN}}} \mathbf{z}_{\text{pre}} \xrightarrow{\text{PQC}} \mathbf{z}_{\text{q}} \xrightarrow{f_{\text{PostDNN}}} \mathbf{y}_{\text{pred}}}.$$

All quantum layer's bit-strings are computed by the CUDA-Q state-vector simulator, and no qtDNN surrogate participates in this inference path.

## C.3 BACKWARD PASS WITH THE QTDNN SURROGATE (SUPERVISED TRAINING, NO RL)

For gradient-based training we keep the forward topology of Sec. C.2 but replace the non-differentiable PQC$\longmapsto \mathbf{z}_{\text{q}}$ map by its qtDNN surrogate, yielding the computational graph

$$\mathbf{x} \xrightarrow{f_{\text{Pre}}} \mathbf{q}_{\text{i}} \xrightarrow{f_{\text{qtDNN}}} \hat{\mathbf{z}}_{\text{q}} \xrightarrow{f_{\text{Post}}} \mathbf{y}_{\text{pred}} \xrightarrow{\mathcal{L}(\,\cdot\,,y_{\text{true}})} \mathcal{L}_{\text{sup}}.$$

The loss $\mathcal{L}_{\text{sup}}$ is task-specific (e.g. MSE or cross-entropy). Because $f_{\text{qtDNN}}$ is an ordinary neural network, automatic differentiation supplies $\nabla_{\theta_{\text{Post}}}\mathcal{L}_{\text{sup}}$, $\nabla_\omega \mathcal{L}_{\text{sup}}$, and $\nabla_{\theta_{\text{Pre}}}\mathcal{L}_{\text{sup}}$ without invoking the quantum simulator. During the same update step we refine the surrogate itself by a second objective, the BCE in Eq. equation 6, computed on the cached pair $(\mathbf{q}_{\text{i}}, \mathbf{z}_{\text{q}})$ from the forward PQC evaluation:

$$\omega \leftarrow \omega - \eta_{\text{qt}} \nabla_\omega \Big( \mathcal{L}_{\text{sup}} + \lambda_{\text{BCE}} \mathcal{L}_{\text{qtDNN}} \Big).$$

Thus the qtDNN plays a dual role: it provides differentiable expectation values for end-to-end learning and simultaneously learns to approximate the true quantum channel, all while keeping the backward pass strictly classical and GPU-resident.

## C.4 REINFORCEMENT LEARNING

The hDQNN-TD3's forward pass with RL is already included in App. D, as Algorithm 2. This section will simply introduce the underlying concepts in higher depth.

### C.4.1 NOTATIONS AND DEFINITIONS

- **State and Action Dimensionality:**

$$s \in \mathbb{R}^{n_s} \quad \text{(the observed state in practice)},$$
$$a \in \mathbb{R}^{n_a} \quad \text{(the action suggested by the actor)}.$$

- **Transitions and Replay Buffer:** At each timestep $t$, we observe $(s_t, a_t, r_t, s_{t+1}, q_i, q_o)$ where $r_t \in \mathbb{R}$ is the reward, and $s_{t+1}$ is the next state governed by $P(s_{t+1} \mid s_t, a_t)$. Each 6-tuple

$$(s, a, r, s', q_{\text{i}}, q_{\text{o}})$$

is stored in a *replay buffer*, $\mathcal{M}$. During training, we sample mini-batches $\{(s, a, r, s', q_{\text{i}}, q_{\text{o}})\}$ from $\mathcal{M}$ to update the actor and critic networks.

### C.4.2 ACTOR-CRITIC FRAMEWORK

- **Critic Model:**

Let $C : (s, a) \mapsto \hat{R}$ be the critic, parameterised by $\theta_C$. It estimates the expected cumulative reward (or return). We define the one-step temporal-difference (TD) target:

$$R^{\text{target}} = r + \gamma\, C\big(s', A(s'; \theta_A)\big),$$

where $\gamma \in (0, 1)$ is the discount factor, and $A(s'; \theta_A)$ is the actor's action in state $s'$. The critic loss is:

$$\mathcal{L}_C(\theta_C) = \mathbb{E}_{(s,a,r,s') \sim \mathcal{M}} \left[ \left( C(s, a; \theta_C) - \left( r + \gamma C(s', A(s'; \theta_A))) \right) \right)^2 \right],$$

where $\mathcal{M}$ is the replay buffer.

- **Actor Model (hDQNN):**
  The actor $A(s; \theta_A)$ is a hybrid deep quantum neural network that outputs an action $a$. In particular, it can be decomposed as follows:

  $$\mathbf{q_i} = f_{\text{preDNN}}(s; \theta_A),$$

  $$\mathbf{q_o} \sim P(\mathbf{q_o} \mid \mathbf{q_i}) = \text{(Measurement of PQC parameterised by } \mathbf{q_i}),$$

  $$a = A(s; \theta_A) = f_{\text{postDNN}}(\mathbf{q_o}, \mathbf{z}_{\text{c-link}}; \theta_A),$$

  where $\mathbf{z}_{\text{c-link}}$ indicates possible direct links from the output of $f_{\text{preDNN}}$ to the input of $f_{\text{postDNN}}$. The distribution $P(\mathbf{q_o} \mid \mathbf{q_i})$ captures the probability of measuring a particular bit-string $\mathbf{q_o}$ on the quantum hardware.

- **Actor Update:**
  To update the parameters $\theta_A$ of the actor model, we often *maximize* the expected return as estimated by the critic:

  $$\mathcal{L}_A(\theta_A) = \mathbb{E}_{s \sim \mathcal{M}} \left[ C(s, A(s; \theta_A); \theta_C) \right].$$

  In practice, gradients $\nabla_{\theta_A} \mathcal{L}_A$ are computed. Here, the qtDNN can be used to approximate the map $\mathbf{q_i} \mapsto \mathbf{q_o}$ locally. This allows batched back-propagation through the classical layers without repeatedly querying the quantum circuit for gradients.

## C.5 Forward pass of hDQNN-TD3 (interaction phase)

At training time the hybrid agent interacts with the environment according to the TD3 loop sketched in Algorithm 1 of the main paper. For each environment step $t$ we execute

1. **Actor evaluation.** The current observation $s_t$ is propagated through the hybrid actor exactly as in Sec. C.2, i.e. $s_t \xrightarrow{\text{PreDNN}} \mathbf{q}_{i,t} \xrightarrow{\text{PQC}} \mathbf{z}_{q,t} \xrightarrow{\text{PostDNN}} a_t$. Exploration noise $\varepsilon_t \sim \mathcal{N}(0, \sigma^2 I)$ is then added:
   $$\tilde{a}_t = \text{clip}(a_t + \varepsilon_t, a_{\min}, a_{\max}).$$

2. **Environment step.** We execute $\tilde{a}_t$ in the environment, observe $r_t$, $s_{t+1}$, $\text{done}_t$ and store the transition $(s_t, \tilde{a}_t, r_t, s_{t+1}, \mathbf{q}_{i,t}, \mathbf{z}_{q,t})$ in the replay buffer $\mathcal{M}$.

Only the genuine PQC is queried during the interaction phase; the qtDNN is *not* used here.

## C.6 Backward pass of hDQNN-TD3 (update phase)

Every $u$ environment steps we draw a mini-batch $\mathcal{B} \subset \mathcal{D}$ and perform the following updates.

**1. Critic update (twice).** For each transition $(s, a, r, s', \mathbf{q_i}, \mathbf{z_q}) \in \mathcal{B}$ compute the bootstrap target

$$y = r + \gamma \min_{j=0,1} C_j^{\text{tar}}(s', a^{\text{tar}}(s')),$$

where the *target* actor–critic pair on $s' = s_{t+1}$:

$$a^{\text{tar}}(s') = A_{\text{tar}}(s') + \text{clip}(\varepsilon', -c, +c), \quad \varepsilon' \sim \mathcal{N}(0, \sigma'^2 I).$$

then minimise the MSE for both critics $C_0, C_1$:

$$\mathcal{L}_{C_j} = \frac{1}{|\mathcal{B}|} \sum_{(s,a) \in \mathcal{B}} (C_j(s, a) - y)^2, \qquad j = 0, 1.$$

**2. Actor surrogate update (delayed).** Every $N_A = 2$ critic steps we update the actor parameters $\theta_A$ following the original TD3:

1. Replace the non-differentiable mapping $\mathbf{q}_i \mapsto \mathbf{z}_q$ by the qtDNN prediction $\hat{\mathbf{z}}_q = f_{\text{qtDNN}}(\mathbf{q}_i; \omega)$.

2. Construct actions $\hat{a} = f_{\text{PostDNN}}(\hat{\mathbf{z}}_q, \cdot)$ and minimise the deterministic TD3 objective

$$\mathcal{L}_A(\theta_A) = -\frac{1}{|\mathcal{B}|} \sum_{s \in \mathcal{B}} C_0(s, \hat{a}(s)).$$

Autograd now flows through the qtDNN, allowing gradients w.r.t. $\theta_{\text{Pre}}$ and $\theta_{\text{Post}}$ without extra PQC calls.

**3. qtDNN fitting.** On the same mini-batch we refine the surrogate by minimising the BCE loss of Eq. 6:

$$\mathcal{L}_{\text{BCE}}(\omega) = \frac{1}{|\mathcal{B}|} \sum_{(\mathbf{q}_i, \mathbf{z}_q) \in \mathcal{B}} \text{BCE}\Big(\mathbf{z}_q, f_{\text{qtDNN}}(\mathbf{q}_i; \omega)\Big).$$

The qtDNN therefore learns on-the-fly to emulate the quantum circuit while simultaneously providing the actor with differentiable expectation values.

**4. Target-network update.** Finally we softly update the target networks with rate $\tau$:

$$\phi^{\text{tar}} \leftarrow \tau \phi + (1 - \tau) \phi^{\text{tar}}, \qquad \phi \in \{\theta_A, \theta_{C_0}, \theta_{C_1}\}.$$

# D ALGORITHMS

## D.1 CONSTRUCTIVE ALGORITHM FOR THE REALISED FUNCTION Q

---

**Algorithm 1** Evaluating the PQC layer $Q$ with $S$ shots, returning per-qubit empirical marginals

---

**Require:** Input $\mathbf{q}_i$ (gate angles / routing), qubit count $N$, depth $M$, shots $S$
1: Prepare $|\psi_0\rangle \leftarrow |0\rangle^{\otimes N}$
2: **for** $\ell = 1$ to $M$ **do**
3:      **for** $i = 1$ to $N$ **do**
4:          apply $R_y(\theta_{\ell,i}^{(y)}) \, R_z(\theta_{\ell,i}^{(z)})$
5:      **end for**
6:      apply CZ on $(p_\ell, k_\ell)$ if $p_\ell \neq k_\ell$
7: **end for**
8: Measure all qubits in $Z$ for $S$ shots $\Rightarrow \{\mathbf{z}^{(s)}\}_{s=1}^{S}$
9: $\hat{p}_i \leftarrow \frac{1}{S} \sum_{s=1}^{S} \mathbb{I}[z_i^{(s)} = 1]$ for $i = 1 \dots N$
10: **return** $\hat{\mathbf{p}} \in [0,1]^N$             ▷ equivalently $\hat{\mathbf{m}} = 1 - 2\hat{\mathbf{p}}$

---

## D.2 TRAINING LOOP PSEUDO-CODE FOR HDQNN-TD3

# E ADDITIONAL GRAPHS AND TABLES

## E.1 MODEL CONFIGURATION

In this table, $\alpha$ is the learning rate and the Critic follows the structure from Lillicrap et al. (2019)

## E.2 EXTENDED ABLATION

Table 3 shows the extended ablation on `Humanoid-v4`. Test returns are averaged over the listed seeds. Because each PQC setting can exceed 24 GPU-hours, we could not complete every random seed for every checkpoint before the submission deadline. Where fewer than five seeds are reported

---

**Algorithm 2** HDQNN-TD3: TD3 actor with a PQC and qtDNN surrogate

---

**Require:** Environment $\mathcal{E}$, replay buffer $\mathcal{M}$, actor $A$ (PQC inside), twin critics $C_1, C_2$, target networks $A'$, $C_1'$, $C_2'$, update periods $N_{\mathrm{A}}$, surrogate epochs $N_{\mathrm{qt}}$
1: **for** episode $= 1, 2, \ldots$ **do**
2:     **collect experience:** interact using $\mathbf{a}_t = A(\mathbf{s}_t) + \mathcal{N}_t$, store $(\mathbf{s}_t, \mathbf{a}_t, \mathbf{q}_i, \mathbf{q}_o, r_t, \mathbf{s}_{t+1})$ in $\mathcal{M}$
3:     **if** time to update **then**
4:         Sample mini-batch $\mathcal{B}$ of size $N_{\mathrm{b}}$ from $\mathcal{M}$
5:         **(a) update qtDNN:** train surrogate for $N_{\mathrm{qt}}$ tiny-batches $\{(\mathbf{q}_i, \mathbf{q}_o)\} \in \mathcal{B}$
6:         **(b) critic step:** TD target $y = r + \gamma \min_{j=1,2} C_j'\big(\mathbf{s}', A'(\mathbf{s}')\big)$; regress $C_1, C_2$ to $y$
7:         **if** global step mod $N_{\mathrm{A}} = 0$ **then**
8:             **(c) actor step:** maximize $C_1(\mathbf{s}, A_{\mathrm{qt}}(\mathbf{s}))$ w.r.t. classical actor parameters
9:             **(d) target update:** $\theta' \leftarrow \tau\theta + (1 - \tau)\theta'$
10:         **end if**
11:     **end if**
12: **end for**

---

Table 2: hDQNN-TD3 Reinforcement Learning Model Configuration for `Humanoid-v4` Problem

| Components | DNN Design | | | | Hyperparameters | | | |
| | Layers | Activation | $N, M$ | $d_{\text{c-link}}$ | $\alpha$ | $\gamma$ | $\tau$ | noise |
|---|---|---|---|---|---|---|---|---|
| PreDNN | [376,256,230] | LReLU | | 10 | $3e^{-4}$ | | $5e^{-3}$ | |
| qtDNN | [220,2 048,10] | LReLU | | | $3e^{-4}$ | | | |
| PQC | | | 10,10 | | | | | $1e^{-3}$ |
| PostDNN | [20,256,17] | LReLU | | 10 | $3e^{-4}$ | | $5e^{-3}$ | $5e^{-2}$ |
| Critic Block 1 | [376,256] | LReLU | | | $3e^{-4}$ | | $5e^{-3}$ | |
| Critic Block 2 | [273,256,1] | LReLU | | | $3e^{-4}$ | 0.99 | $5e^{-3}$ | |

(see the "Seeds" column), the missing runs were still in progress; the completed runs are shown to provide early insight rather than to claim definitive rankings.

For coherence purpose, we also display Table 4, which shows the extended ablation on `Humanoid-v4` with seed 3 omitted, as it couldn't run further than 6200 episodes before the submission deadline.

### E.3 DETAILED GRAPHS WITH SEEDS

Figure 6 juxtaposes four learning curves obtained from an ablation study on the quantum layer's hyper-parameters. Each subplot displays the 100-episode moving-average return on `Humanoid-v4` produced by the hDQNN-TD3 backbone while varying both the number of circuit shots, $\{100, 1000\}$, and the number of active qubits, $\{5, 10\}$. Every solid, colored trace corresponds to a single random seed; no additional smoothing or confidence band is applied, so the visible oscillations are the raw training dynamics. The comparison indicates that (i) increasing the qubit count from five to ten accelerates early learning only when the shot budget remains low, whereas (ii) allocating more shots generally stabilizes convergence but yields high returns chiefly in the smaller-qubit regime.

### E.4 OTHER SUPPLEMENTARY FIGURES

## F TECHNICAL STACK

### F.1 HARDWARE AND SYSTEM SOFTWARE.

All numerical experiments were executed independently on a dedicated NVIDIA A100-SXM4 GPU (40 GB HBM2e). The software stack comprised:

- `Python 3.12.3;`

Table 3: Extended ablation on `Humanoid-v4`. Test returns are averaged over $4 - 5$ seeds; the right-most column lists the seed indices contributing to each mean.

| Ep. | Layer | Shots | Qu. | Mean ret. | Best ret. | Time | PQC calls | Seeds |
|---|---|---|---|---|---|---|---|---|
| 4 000 | FC | 0 | 0 | $423.0 \pm 170.8$ | 603.6 | 31.5 | 214 686 | 0,1,2,3,4 |
| 4 000 | RBG | 0 | 0 | $333.6 \pm 196.5$ | 545.3 | 35.7 | 198 449 | 0,1,2,3,4 |
| 4 000 | PQC | 100 | 5 | $1\,575.9 \pm 1\,923.4$ | 4 882.7 | 515.2 | 540 138 | 0,1,2,3,4 |
| 4 000 | PQC | 1 000 | 5 | $476.5 \pm 149.4$ | 650.8 | 225.3 | 280 476 | 0,1,2,3,4 |
| 4 000 | PQC | 100 | 10 | $667.3 \pm 442.1$ | 1 453.3 | 315.2 | 335 815 | 0,1,2,3,4 |
| 4 000 | PQC | 1 000 | 10 | $556.0 \pm 81.7$ | 689.4 | 265.5 | 308 709 | 0,1,2,3,4 |
| 6 200 | FC | 0 | 0 | $543.4 \pm 243.3$ | 909.4 | 72.2 | 423 010 | 0,1,2,3,4 |
| 6 200 | RBG | 0 | 0 | $417.0 \pm 131.4$ | 526.2 | 75.5 | 366 904 | 0,1,2,3,4 |
| 6 200 | PQC | 100 | 5 | $2\,014.7 \pm 2\,008.6$ | 4 342.4 | 1 596.6 | 1 387 705 | 0,1,2,3,4 |
| 6 200 | PQC | 1 000 | 5 | $544.8 \pm 395.1$ | 1 102.7 | 467.0 | 517 611 | 0,1,2,3,4 |
| 6 200 | PQC | 100 | 10 | $1\,492.3 \pm 2\,191.2$ | 5 411.2 | 988.9 | 959 152 | 0,1,2,3,4 |
| 6 200 | PQC | 1 000 | 10 | $657.0 \pm 145.0$ | 841.6 | 552.6 | 610 225 | 0,1,2,3,4 |
| 7 000 | FC | 0 | 0 | $693.3 \pm 453.1$ | 1 473.0 | 98.5 | 515 399 | 0,1,2,3,4 |
| 7 000 | RBG | 0 | 0 | $466.7 \pm 106.0$ | 547.2 | 92.8 | 438 296 | 0,1,2,3,4 |
| 7 000 | PQC | 100 | 5 | $2\,139.6 \pm 2\,180.2$ | 4 962.0 | 1 935.9 | 1 756 266 | 0,1,2,3,4 |
| 7 000 | PQC | 1 000 | 5 | $733.1 \pm 740.4$ | 1 974.3 | 612.5 | 642 867 | 0,1,2,3,4 |
| 7 000 | PQC | 100 | 10 | $532.7 \pm 105.6$ | 690.6 | 678.9 | 666 056 | 0,1,2,4 |
| 7 000 | PQC | 1 000 | 10 | $710.1 \pm 238.6$ | 1 040.6 | 675.5 | 744 615 | 0,1,2,3,4 |
| 7 200 | FC | 0 | 0 | $617.5 \pm 341.1$ | 1 207.8 | 108.8 | 544 790 | 0,1,2,3,4 |
| 7 200 | RBG | 0 | 0 | $414.3 \pm 103.0$ | 545.0 | 99.6 | 456 050 | 0,1,2,3,4 |
| 7 200 | PQC | 100 | 5 | $1\,605.0 \pm 2\,202.2$ | 4 906.6 | 1 791.9 | 1 527 037 | 0,1,2,4 |
| 7 200 | PQC | 1 000 | 5 | $1\,078.5 \pm 1\,517.3$ | 3 759.2 | 673.0 | 692 138 | 0,1,2,3,4 |
| 7 200 | PQC | 100 | 10 | $517.8 \pm 95.0$ | 659.2 | 712.2 | 694 104 | 0,1,2,4 |
| 7 200 | PQC | 1 000 | 10 | $763.5 \pm 335.7$ | 1 277.4 | 711.9 | 782 646 | 0,1,2,3,4 |
| 7 700 | FC | 0 | 0 | $1\,126.1 \pm 1\,467.2$ | 3 745.5 | 135.4 | 621 311 | 0,1,2,3,4 |
| 7 700 | RBG | 0 | 0 | $432.4 \pm 97.4$ | 543.2 | 110.8 | 500 487 | 0,1,2,3,4 |
| 7 700 | PQC | 100 | 5 | $1\,402.8 \pm 1\,757.3$ | 4 031.1 | 1 943.2 | 1 688 888 | 0,1,2,4 |
| 7 700 | PQC | 1 000 | 5 | $1\,121.4 \pm 1\,558.9$ | 3 868.6 | 839.3 | 824 076 | 0,1,2,3,4 |
| 7 700 | PQC | 100 | 10 | $506.6 \pm 97.0$ | 624.5 | 793.6 | 765 043 | 0,1,2,4 |
| 7 700 | PQC | 1 000 | 10 | $891.8 \pm 547.1$ | 1 852.9 | 805.0 | 882 668 | 0,1,2,3,4 |

- `PyTorch 2.6.0` built against `CUDA 12.4.0`;

- `Gymnasium 1.1.1` with `MuJoCo 3.3.2`;

- `cuda-q 0.9.1` for GPU-accelerated quantum-circuit simulation NVIDIA (2022).

## F.2   NVIDIA CUDA-Q

We adopted CUDA-Q[7] as our quantum-simulation engine for one main reason: the GPU is a light weight and agile way to develop and carry out small-scale tests for new model architectures with a small number of qubits involved. Fine-grained quantitative investigations could be carried out easily in this way to understand the behaviours of novel hybrid quantum-classical machine learning architectures by limiting the qubit count, offering direct insights into their foundation of learning and information processing flows. CUDA-Q provides GPU-accelerated tensor-network and state-vector simulators for quick proof-of-concept demonstrations on GPUs. The hDQNN architectures validated in through this could be straightforwardly scaled towards beyond classical large models (with a medium or large qubit count) running on hybrid systems having classical computing resources (CPU/GPU) interconnected with physical (noisy) QPUs.

---

[7]https://developer.nvidia.com/cuda-quantum

Table 4: Ablation on `Humanoid-v4` with seed 3 still running. Each mean is averaged over the seeds listed in the right-most column.

| Ep. | Layer | Shots | Qu. | Mean ret. | Best ret. | Time | PQC calls | Seeds |
|---|---|---|---|---|---|---|---|---|
| 4 000 | FC | 0 | 0 | $382.7 \pm 167.5$ | 603.6 | 27.4 | 196 401 | 0,1,2,4 |
| 4 000 | RBG | 0 | 0 | $356.5 \pm 219.1$ | 545.3 | 39.8 | 210 584 | 0,1,2,4 |
| 4 000 | PQC | 100 | 5 | $1\,555.4 \pm 1\,923.4$ | 4 882.7 | 564.8 | 574 890 | 0,1,2,4 |
| 4 000 | PQC | 1 000 | 5 | $521.3 \pm 128.0$ | 650.8 | 237.5 | 302 738 | 0,1,2,4 |
| 4 000 | PQC | 100 | 10 | $470.8 \pm 57.1$ | 534.5 | 291.2 | 302 948 | 0,1,2,4 |
| 4 000 | PQC | 1 000 | 10 | $568.0 \pm 89.2$ | 689.4 | 258.4 | 303 057 | 0,1,2,4 |
| 6 200 | RBG | 0 | 0 | $389.7 \pm 134.3$ | 521.2 | 80.4 | 372 226 | 0,1,2,4 |
| 6 200 | FC | 0 | 0 | $514.3 \pm 270.8$ | 909.4 | 61.1 | 384 622 | 0,1,2,4 |
| 6 200 | PQC | 100 | 5 | $1\,432.7 \pm 1\,766.9$ | 4 075.3 | 1 489.3 | 1 196 503 | 0,1,2,4 |
| 6 200 | PQC | 1 000 | 5 | $602.9 \pm 430.8$ | 1 102.7 | 498.4 | 567 036 | 0,1,2,4 |
| 6 200 | PQC | 100 | 10 | $512.5 \pm 52.6$ | 590.5 | 551.2 | 557 415 | 0,1,2,4 |
| 6 200 | PQC | 1 000 | 10 | $652.4 \pm 167.0$ | 841.6 | 544.0 | 593 357 | 0,1,2,4 |
| 7 000 | FC | 0 | 0 | $691.6 \pm 523.2$ | 1 473.0 | 81.9 | 467 999 | 0,1,2,4 |
| 7 000 | RBG | 0 | 0 | $446.5 \pm 110.8$ | 543.4 | 98.2 | 440 609 | 0,1,2,4 |
| 7 000 | PQC | 100 | 5 | $1\,666.9 \pm 2\,201.8$ | 4 962.0 | 1 730.9 | 1 460 228 | 0,1,2,4 |
| 7 000 | PQC | 1 000 | 5 | $837.8 \pm 811.1$ | 1 974.3 | 664.2 | 711 615 | 0,1,2,4 |
| 7 000 | PQC | 100 | 10 | $532.7 \pm 105.6$ | 690.6 | 678.9 | 666 056 | 0,1,2,4 |
| 7 000 | PQC | 1 000 | 10 | $698.3 \pm 273.8$ | 1 040.6 | 665.6 | 724 890 | 0,1,2,4 |
| 7 200 | FC | 0 | 0 | $620.8 \pm 393.8$ | 1 207.8 | 89.8 | 494 098 | 0,1,2,4 |
| 7 200 | RBG | 0 | 0 | $426.8 \pm 114.5$ | 545.0 | 105.9 | 458 399 | 0,1,2,4 |
| 7 200 | PQC | 100 | 5 | $1\,605.0 \pm 2\,202.2$ | 4 906.6 | 1 791.9 | 1 527 037 | 0,1,2,4 |
| 7 200 | PQC | 1 000 | 5 | $1\,269.5 \pm 1\,681.1$ | 3 759.2 | 735.6 | 770 278 | 0,1,2,4 |
| 7 200 | PQC | 100 | 10 | $517.8 \pm 95.0$ | 659.2 | 712.2 | 694 104 | 0,1,2,4 |
| 7 200 | PQC | 1 000 | 10 | $768.9 \pm 387.4$ | 1 277.4 | 700.9 | 761 498 | 0,1,2,4 |
| 7 700 | FC | 0 | 0 | $1\,253.1 \pm 1\,662.1$ | 3 745.5 | 115.1 | 573 588 | 0,1,2,4 |
| 7 700 | RBG | 0 | 0 | $431.2 \pm 112.4$ | 543.2 | 118.3 | 504 233 | 0,1,2,4 |
| 7 700 | PQC | 100 | 5 | $1\,402.8 \pm 1\,757.3$ | 4 031.1 | 1 943.2 | 1 688 888 | 0,1,2,4 |
| 7 700 | PQC | 1 000 | 5 | $1\,326.7 \pm 1\,720.3$ | 3 868.6 | 933.4 | 927 753 | 0,1,2,4 |
| 7 700 | PQC | 100 | 10 | $506.6 \pm 97.0$ | 624.5 | 793.6 | 765 043 | 0,1,2,4 |
| 7 700 | PQC | 1 000 | 10 | $945.3 \pm 616.5$ | 1 852.9 | 788.8 | 855 506 | 0,1,2,4 |

During the proof-of-concept trainings on GPU, every forward pass of the PQC is dispatched to the CUDA-Q state-vector simulator in single precision (FP32). The PQC is executed repeatedly with the same parameters over multiple shots. The most probable bit-string from measuring the qubits in the final step of each shot is passed back from the quantum kernel as the output of the quantum layer (in a PQC Call as defined in the main text) to `PostDNN`. In this process, all classical DNNs and the connected quantum kernel are hosted on one GPU node.

### F.3 RESOURCE FOOTPRINT OF THE SURROGATE

A 10-qubit qtDNN has 473,098 trainable parameters ($\approx 1.80\,\text{MB}$ of FP32 weights). Adam adds two moments of equal size ($\approx 3.60\,\text{MB}$), keeping the persistent footprint under 6 MB. On our A100 40GB runs, qtDNN updates account for $\sim 2\%$ of wall-clock (median across seeds).

### F.4 DEPLOYMENT REGIME

All experiments execute the PQC with CUDA-Q's shot-based simulator augmented by bit/phase-flip noise channels to emulate NISQ behaviour. At test time the policy executes the *same* noisy PQC (on calibrated hardware or its simulator). The qtDNN is never used for inference.

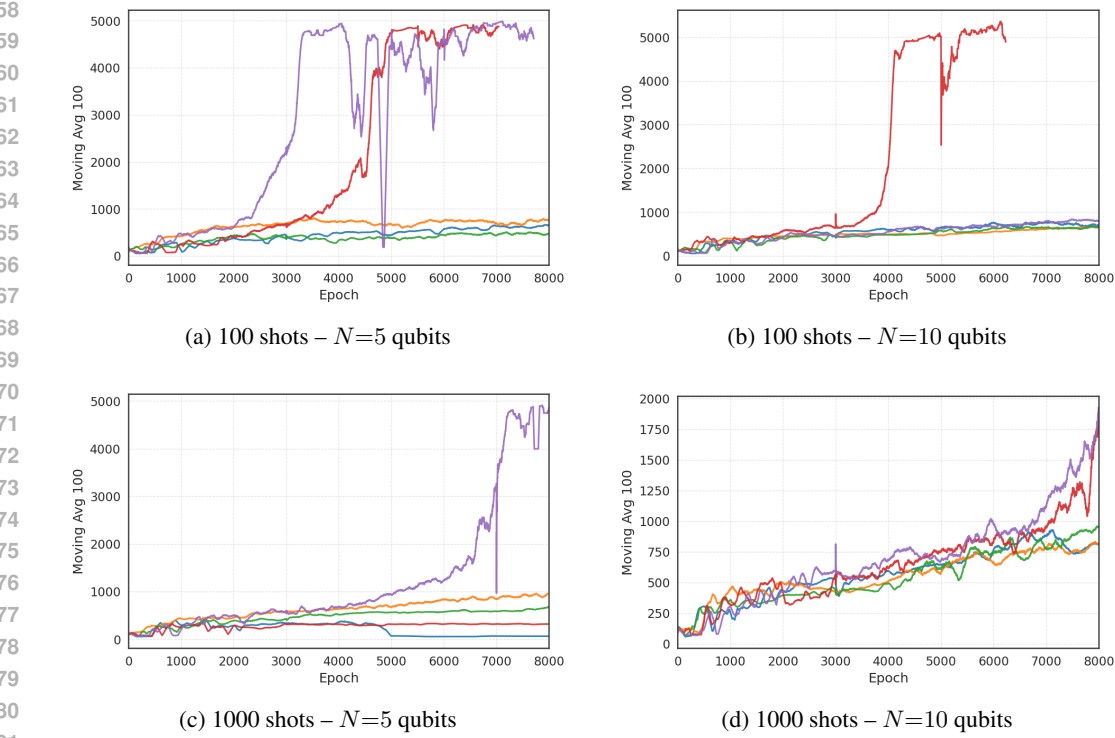

(a) 100 shots – $N=5$ qubits

(b) 100 shots – $N=10$ qubits

(c) 1000 shots – $N=5$ qubits

(d) 1000 shots – $N=10$ qubits

Figure 6: Moving-average (100-episode window) return curves for four PQC configurations. Each colored trace corresponds to one random seed, consistent across the experiments.

### F.5 NOISE SOURCES IN NISQ-COMPATIBLE TRAINING

Throughout training, the system operates under three distinct noise regimes:

- **Sampling noise**, arising from finite-shot PQC measurements, which produce stochastic bit-strings even in the absence of gate errors (we considered 100 and 1000 shots in the ablation study).
- **Gate noise**, stemming from hardware-level imperfections and is modeled via bit-flip and phase-flip channels with realistic error rates ($\sim 10^{-3}$ in the ablation study).
- **Exploration noise**, entering through stochastic policy perturbations used during environment interaction via application of Gaussian noise.

The qtDNN surrogate is trained directly on the resulting noisy bit-strings and thus inherits robustness to all three sources during gradient-based optimisation.

## G QUANTUM ADVANTAGE OF PQC OVER CLASSICAL ARCHITECTURES

### G.1 COMPACT MULTI-FREQUENCY STRUCTURE.

Expectation values of data-encoded PQCs admit a finite Fourier series; the largest frequency scales at most linearly with circuit depth. A shallow ReLU layer can match the same spectra only by increasing width rapidly with frequency and dimension, implying an exponential width gap in the multi-dimensional setting (cf. Schuld et al. (2021); Montúfar et al. (2014)).

### G.2 ENTANGLEMENT AS AN INDUCTIVE BIAS.

The entangling feature map induces a data-dependent quantum kernel that captures non-local correlations which comparable-size random-feature maps struggle to reproduce; when the geometric

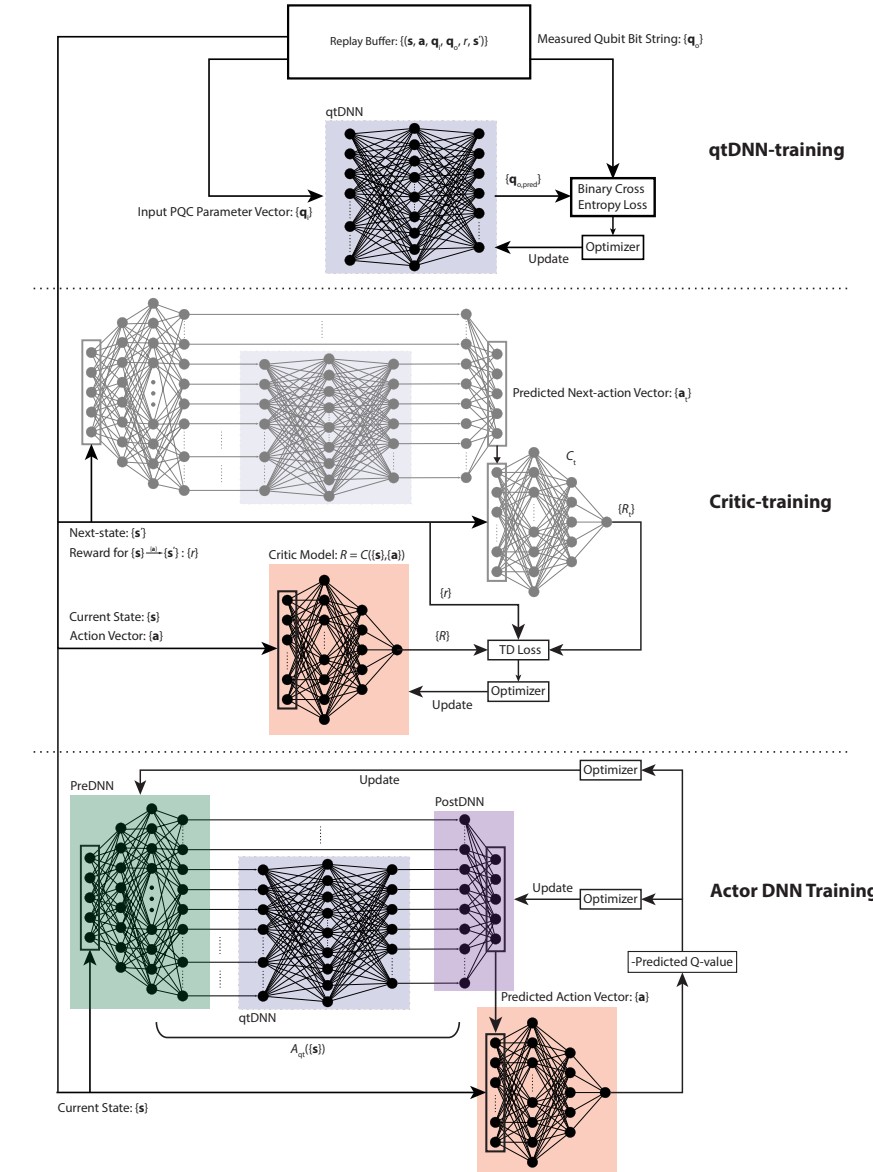

Figure 7: The batched training process flow that incorporates qtDNN training framework for efficiently updating the classical neural layers of the hDQNN model to achieve desired learning goals.

discrepancy is large, one can construct datasets on which the quantum model achieves strictly lower prediction error (cf. Huang et al. (2021b)).

Empirically, replacing the PQC with an FC layer of identical width lowers mean return by $+296$ at 4k episodes and $+148$ at 7.2k episodes (Table 2), consistent with these lenses.

