# OpenReview forum: "Training Hybrid Deep Quantum Neural Network for Efficient Reinforcement Learning"
_ICLR.cc/2026/Conference — ICLR 2026 Conference Withdrawn Submission_

### Official Review · Reviewer_VpTy · 2025-10-29

**Soundness:** 3
**Presentation:** 3
**Contribution:** 3
**Rating:** 2
**Confidence:** 4

**Summary:**

The paper focuses on enhancing the training of so called hDQNNs reinforcement learning models which are characterized by a combination of PQC and neural network components. In their new model they utilize a small surrogate network called qtDNN which learns PQC components in minibatches by multi-layered perceptrons (the quality of surrogation is controlled via a theorem) enabling more efficient backward passes.

The method is instantiated in the hDQNN-TD3 architedture (an actor-critic set up) and tested in the Humanoid-v4 benchmark setting, and includes settings with 5 and 10 qubits in the PQC layer, and the use of 100 and 1000 shots.
The numerics include ablation studies, for me the most interesting being the one which showcases that the PQC has a functional role.

**Strengths:**

A theoretically supported mechanism to speed up training of hybrid QRL is introduced. The performance of the model is nicely benchmarked and very strong performacne is obtained (but not anywhere close to a quatum advantage regime so any discussion of this is not supportable).
Clean presentation.

**Weaknesses:**

The theory seems not to say much about what MLP should be used nor how efficient the learning will be.
The role of PQCs for these tasks is quite unclear.
 The RBG method is attaining very strong results comparatively.
I would be worried some more hyperparameter optimization could show PQC quite useles.. but I agree I have no evidence for this.
I would be much more convinced if the model was trained on some problem where there is an a-priori reason that quantum models could be advantageous.
Maybe I misunderstood, but did 5 qubit runs outperform 10 qubit runs? This would be a sign of no advantage.
In general, without some pathway for quantum advantage being discussed I cannot see the paper being of high interest for a broad community.

**Questions:**

1) does theory say anything about what kind of MLP should be used and about sample complexity for learning
2) if the surrogate method works for RL, why would it not work for supervised learning, or more generally for VQEs? This is an extremely important problem (training) and the use of surrogates in SL and VQEs would remove a lot of "noise" coming from the complicated RL setting to see how well it works.
3) How would all this scale? What happens when the circuits hit vanishing gradients?

---

### Official Review · Reviewer_5wnM · 2025-10-31

**Soundness:** 2
**Presentation:** 2
**Contribution:** 2
**Rating:** 4
**Confidence:** 3

**Summary:**

This paper addresses the critical bottleneck of gradient estimation in training hybrid deep quantum neural networks (hDQNNs), particularly within reinforcement learning (RL). Current methods, such as the parameter-shift rule, are computationally prohibitive for realistic workloads as they require an intractable number of quantum circuit evaluations and do not support parallel mini-batch processing.

The authors introduce qtDNN. The qtDNN is a classical, differentiable "tangential surrogate" network that is trained on-the-fly to locally approximate the Parameterised Quantum Circuit (PQC) within each training mini-batch. Crucially, this surrogate is only used during the backward pass to provide scalable, GPU-friendly gradients. The actual PQC remains in the computation graph for the forward pass during inference and environment interaction, thus preserving the quantum model's expressive power.

**Strengths:**

The method is grounded by Theorem 1 (Local Gradient Fidelity), which leverages universal approximation to guarantee that an MLP (the qtDNN) can approximate both the PQC's output ($Q(x)$) and its Jacobian ($\nabla Q(x)$) within an arbitrary tolerance in a local region (the mini-batch). This provides a justification for using the surrogate's gradient in the backward pass.

The experimental setup is excellent. The choice of Humanoid-v4  as the primary benchmark is ambitious and moves beyond the "toy" problems often seen in QRL. The authors compare against strong, publicly-available classical baselines, ensuring a fair and meaningful comparison. The ablation study (PQC vs. FC vs. RBG)  is critical and effectively isolates the contribution of the quantum layer.

**Weaknesses:**

- The most significant weakness is the exponential scaling of the qtDNN surrogate itself, which is discussed in Appendix A.1.1. The authors note that their 3-layer MLP surrogate requires a hidden layer width of $\mathcal{L}_h \propto 2^{N+1}$ neurons to be effective. This implies that the classical part of the training (the surrogate update) has a memory and computational cost that scales exponentially with the number of qubits ($N$). This trades one form of exponential bottleneck (parameter-shift evaluations) for another (classical surrogate complexity). This limitation is critical to the method's scalability in $N$ and should be discussed more prominently in the main paper, not just in the appendix.

- Related to the first point, the choice of a generic 3-layer MLP as the surrogate  is likely parameter-inefficient. A more structured classical network (e.g., a Tensor Network, as the authors suggest in future work ) that mimics the PQC's entanglement structure might achieve the same gradient fidelity with far fewer parameters, potentially breaking the exponential scaling.

- The ablation study proves the PQC layer is beneficial, but the reason for this benefit (App. G)  is speculative. The paper suggests entanglement and multi-frequency spectra as a post-hoc explanation. A deeper investigation into the feature space learned by the PQC (e.g., via representation analysis) versus the FC layer would be needed to make a stronger claim about the nature of the quantum advantage.

- The method's stability relies on the qtDNN being successfully re-fit at every actor update step (using $N_{qt}=32$ tiny-batches)3131. The paper shows the BCE loss is stable (Fig. 3c), but it lacks a sensitivity analysis on $N_{qt}$ or the surrogate's width $L_h$. How much does the policy's final performance degrade if the surrogate fit is mediocre (e.g., if $N_{qt}$ is reduced to 1 or 5)? This would help establish the method's robustness.

**Questions:**

1. Exponential Surrogate Scaling: The Appendix (A.1.1) states the qtDNN hidden layer $L_h$ scales as $2^{N+1}$. This seems to be the method's primary limitation, restricting it to small $N$ (e.g., $N=20$ qubits would require a hidden layer of $\approx 2M$ neurons, consuming $\approx 8TB$ of memory for a 20-dim output, which is infeasible). Do the authors believe this exponential scaling is fundamental to locally approximating a PQC's gradient with an MLP, or is it an artifact of the simple MLP architecture? How does this scaling impact the "path toward large-scale RL" claim?

2. Theorem 1 assumes the PQC is a deterministic, smooth map $Q \in C^1$. However, the qtDNN is trained on $q_o$, which is the noisy, finite-shot empirical marginal vector. This means the surrogate is approximating a stochastic function. How does this noise affect the gradient fidelity guarantees of Theorem 1? Does the $\epsilon_2$ bound on the gradient error still hold, or does it become a guarantee on the expectation?

3. The surrogate is trained with $N_{qt}=32$ updates on tiny-batches drawn from the same mini-batch $\mathcal{B}$. This seems to risk overfitting the surrogate to that specific mini-batch. What was the reasoning for $N_{qt}=32$? Have the authors explored the trade-off between a better local fit (high $N_{qt}$) and the risk of overfitting to the batch, which might provide poor gradients for the underlying local function?

4. The results on Humanoid-v4 are excellent. However, both benchmarks tested (Humanoid, Hopper) are MuJoCo locomotion tasks. How do the authors hypothesize this method would perform on RL tasks with different characteristics, such as tasks with high-dimensional visual inputs (which would require a large CNN PreDNN) or tasks with very sparse rewards?

---

### Official Review · Reviewer_SNua · 2025-10-31

**Soundness:** 2
**Presentation:** 2
**Contribution:** 2
**Rating:** 2
**Confidence:** 5

**Summary:**

In this work, the authors propose hDQNN-TD3, a hybrid quantum-classical actor-critic architecture that introduces a differentiable surrogate module, qtDNN, to approximate PQCs. The authors claim that the qtDNN can locally fit the input-output behavior of the quantum layer in each mini-batch, enabling gradient-based updates of the surrounding classical networks without performing parameter-shift or finite-difference gradient evaluations on realistic quantum hardware. Their experiments on continuous-control tasks show moderate performance gains over classical TD3, SAC, and PPO baselines.

**Strengths:**

1. The hDQNN-TD3 framework provides gradient estimation in PQCs for hybrid RL training.

2. The authors' empirical section demonstrates that the surrogate can stabilize the RL training without explicit quantum gradients.

3. The experiments acknowledge noise and shot limitations on NISQ hardware.

**Weaknesses:**

1. The novelty of this work is limited, and both theoretical and empirical contributions are minimal. As for the quantum surrogate gradient models (e.g., Chen et al., 2023; Jerbi et al., 2021), they already approximate PQCs using differentiable neural networks for hybrid optimization. Moreover, the finite-difference emulators or surrogate quantum layers have been discussed in the following papers.

[1] “Classical surrogates for quantum learning models” (Schreiber, Eisert, Meyer, 2022)

[2] “Surrogate-based optimization for variational quantum algorithms” (Shaffer et al., 2023)

[3] “Emulating quantum dynamics with neural networks via knowledge distillation” (Yao et al., 2022)

2. Theorem 1 merely states that a feed-forward network can approximate the PQC and its gradient within small tolerances, which is a restatement of universal approximation, without proof of convergence, stability, or bounds on ε in terms of circuit depth or shot noise. Furthermore, there is no demonstration that the qtDNN preserves unbiased policy gradients or that it converges to the true quantum gradient distribution.

3. Reported gains on Humanoid-v4 (≈ +148 mean return over TD3) fall within run-to-run variance, and the paper never performs statistical significance tests across seeds or hardware noise settings.

4. The paper provides no evidence of non-classical advantage, entanglement analysis, or scaling beyond 10 qubits. Because the quantum layer is bypassed during training and only used for inference, the model’s improved learning curve stems mainly from additional classical capacity, not from quantum correlations.

5. The Discussion section speculates about “hDQNN-LLM” and “NVIDIA robot controllers,” which is far outside the paper’s evidence base.

**Questions:**

1. Can you explicitly contrast qtDNN with existing surrogate-gradient and emulator methods in quantum RL (Chen et al. 2023; Jerbi et al. 2021)?

2. Is there any experiment on real quantum hardware to justify claims of “hardware-efficiency”?

3. Could you provide quantitative bounds or empirical measures of the ε₁, ε₂ tolerances in Theorem 1?

4. How do you ensure that training on a classical qtDNN does not destroy any quantum-specific representational benefit?

---

### Official Review · Reviewer_LCfb · 2025-11-01

**Soundness:** 3
**Presentation:** 3
**Contribution:** 4
**Rating:** 4
**Confidence:** 3

**Summary:**

This paper proposes qtDNN, a novel learning technique for efficiently applying hybrid quantum-classical neural networks (hDQNNs) to reinforcement learning (RL). The method addresses the computational bottleneck of gradient calculations in quantum circuits, paving the way for the practical integration of quantum layers into deep reinforcement learning models. The authors applied this qtDNN technique to a robust RL algorithm called TD3 to design the hDQNN-TD3 agent, which achieves performance on par with or above that of SOTA classical models on high-level benchmarks such as Humanoid-v4.

**Strengths:**

It is a very original approach to separate the backpropagation process by introducing a classical surrogate model (surrogate) to solve the problem that traditional methods relying on the Parameters-Shift Rule (PSR) do not enable GPU parallelization, requiring enormous training time.

By validating the performance of the proposed technique on highly challenging and complex high-dimensional continuous control benchmarks, we demonstrate that it is sufficiently robust for real-world learning.

**Weaknesses:**

The implementation of a 3-layer MLP of qtDNN requires $2^{N+1}$ neurons for $N$ qubits, and as the number of qubits increases (as the size of the model increases), the qtDNN becomes exponentially huge for the number of qubits. If N is a larger number, there will be limitations in learning qtDNN, but discussions about this are insufficient.

This may be nothing more than shifting the time bottleneck of PSR to the memory bottleneck of qtDNN, and the abstract's claim that it aims to be applied to large-scale RL is overly ambitious.

This paper is very unfriendly to the general ICLR readership, not QML experts. There is a lack of description of why we do not use pure QRL models and the need for hybrid structures.

**Questions:**

How can this exponential classical memory requirement be addressed when the number of qubits increases above $N=20,30$?

What grounds do you think the proposed research can be applied to large-scale RL?

It is necessary to explain the fundamental bottlenecks faced by pure QRL models (i.e., using only PQC) when dealing with high-dimensional continuous input and output, and the differences caused by the proposed techniques.

One of the key motivations for using PQC is to model correlations through entanglement. However, this paper assumes inter-qubit independence. The proposed technique trains a surrogate model with loss functions that ignore correlations, are you sure there are no contradictions and defects in this area? It needs to be explained how it can be delivered correctly to the backpropagation process.

---

### Note · Authors · 2026-01-23

I have read and agree with the venue's withdrawal policy on behalf of myself and my co-authors.